

# Lefschetz thimble-inspired weight regularizations for complex Langevin simulations

**Kirill Boguslavski, Paul Hotzy⋆ and David I. Müller**

Institute for Theoretical Physics, TU Wien, Vienna, Austria

⋆ paul.hotzy@tuwien.ac.at

## Abstract

Complex Langevin (CL) is a computational method to circumvent the numerical sign problem with applications in finite-density quantum chromodynamics and the real-time dynamics of quantum field theories. It has long been known that, depending on the simulated system, CL does not always converge correctly. In this work, we provide numerical evidence that the success or failure of the complex Langevin method is deeply tied to the Lefschetz thimble structure of the simulated system. This is demonstrated by constructing weight function regularizations that deform the thimbles of systems with compact domains. Our results indicate that CL converges correctly when the regularized system exhibits a single relevant compact thimble. We introduce a bias correction to retrieve the values of the original theory for parameter sets where a direct complex Langevin approach fails. The effectiveness of this method is illustrated using several toy models, including the cosine model and the SU(2) and SU(3) Polyakov chains. Finally, we discuss the opportunities and limitations of this regularization approach for lattice field theories.



# 1 Introduction

Lattice field theory is an indispensable tool in both high-energy and condensed-matter physics. It provides a unique framework for non-perturbative calculations of observables through Monte Carlo (MC) simulations. Lattice calculations are applicable to systems in thermal equilibrium, that can be formulated solely using Euclidean time, where the path integral formalism leads to a positive-definite weight function. In systems at finite density or systems involving real-time dynamics, the lattice path integral is generally characterized by non-positive or even complex weight functions. This leads to a long-standing computational challenge, the numerical sign problem [1, 2], that renders traditional MC approaches ineffective. The difficulty of the sign problem is underlined by the fact that it has been shown to be NP-hard [3], suggesting that a general and efficient solution is unlikely to exist.

A number of alternative approaches to traditional Monte Carlo have been proposed in order to circumvent or at least soften the sign problem. Each of these approaches is effective in certain systems, but they also exhibit specific limitations. For quantum chromodynamics (QCD) at finite density, that is at finite baryon chemical potential $\mu$, multiple methods have been devised, among them sophisticated reweighting techniques [4], Taylor expansion around $\mu = 0$ [5, 6], analytic continuation of imaginary chemical potential simulations [7–9], and simulations using canonical ensembles [10, 11]. Obtaining the real-time dynamics of the system, in principle, requires analytically continuing the Euclidean data that is accessible for Lattice QCD. This task can be formulated as an inverse problem, which is often characterized as being ill-posed. Nevertheless, approaches such as spectral reconstruction allowed the calculation of transport coefficients based on lattice data [12–14]. Other important techniques are meron cluster algorithms for fermionic systems [15], tensor networks [16–19], and the density of states method [20, 21]. These approaches have produced remarkable results, but they are typically limited in the range of $\mu$, the lattice size, or they exhibit difficulties in controlling statistical errors.

A different approach to the sign problem, specifically when the weight function becomes complex, is to extend the measure of integration and the degrees of freedom into the complex plane. There are several methods implementing this idea; most notably, the Lefschetz thimble method and contour deformation [22–26], Complex Langevin (CL) [27,28], and combinations thereof [29–31]. Contour deformation techniques are based on changing the integration contour of the path integral of the theory such that the oscillatory nature of the non-positive weight function is strongly reduced. Moreover, these approaches are theoretically appealing because integration over complex contours has a firm mathematical foundation in Picard-Lefschetz theory. These methods have produced state-of-the-art results for many systems [32–36], albeit often only for effective models at small lattice sizes and a low number of spatial dimensions. In practical applications, deforming the path integral contour in four-dimensional field theories presents significant computational challenges.

Complex Langevin, on the other hand, can be considered as the analytical continuation of stochastic quantization [37]. Euclidean stochastic quantization is based on the fact that, under quite general assumptions, the path integral can be approximated by the late-time solution to a Langevin process. In the case of systems with a sign problem due to a complex action, the relevant degrees of freedom are complexified, and the process explores the complex plane. It has been demonstrated for specific systems, this complex stochastic evolution reproduces the correct expectation values of observables. Moreover, CL is easy to formulate for high-dimensional systems and is computationally efficient. Despite as of yet unresolved practical and formal issues, CL has been successfully applied to many relevant systems such as 0+1D and 1+1D real-time scalar fields [38–42], 4D finite density scalar fields [43, 44], 4D heavy-dense quantum chromodynamics [45] and full QCD at finite chemical potential [46–49]. More recently, CL simulations of real-time 3+1D non-abelian gauge theory, which have previously been found to be particularly difficult [50,51], have seen significant improvement through the use of a stabilizing kernel [52,53].

In recent years, significant progress has been made toward diagnosing incorrect convergence in the CL method, with the development of criteria for correctness [28, 54–57] and studies on the role of boundary terms linked to incorrect convergence [58–60]. However, these criteria are generally applied a posteriori, that is, they can only be checked after running numerical simulations. Moreover, they offer limited guidance on how to proceed when they are not satisfied. The main challenge for the CL method is, therefore, twofold: first, we lack a clear understanding of which features of the considered system lead to convergence failure, and second, there is an absence of a constructive strategy for achieving reliable convergence for specific applications of CL. This underscores the need for further research to develop tools that not only diagnose but also provide practical ways to correct or adapt CL for broader applications.

There has been a renewed interest in so-called kernels, which yield different representations of the CL equations. Kernelled CL simulations can exhibit drastically improved stability and convergence properties. Although kernels have been investigated early on in the development of CL [61, 62], the kernel approach has only recently been applied to lattice field theories [63]. In particular, kernels have enabled simulations of real-time scalar fields [40–42] and non-Abelian gauge theories [52,53]. Despite these successes, and in contrast to other formal aspects of CL such as convergence criteria, it appears that kernels are barely understood from a formal viewpoint. As such, the choice of a suitable kernel remains entirely non-trivial.

There are interesting connections between the thimble structure of a theory and the convergence properties of CL. Empirically, it has been observed that CL simulations may naturally sample configurations close to the Lefschetz thimbles and critical points of the action [64,65]. Adding kernels can further influence the stochastic process, steering it closer to these critical regions and the relevant thimbles that contribute to the path integral [40,61]. However, when

multiple relevant thimbles are present, the CL method is likely to fail, as it tends to incorrectly weigh the individual thimble contributions. While in simple models this incorrect weighting can be corrected through reweighting methods [66], this becomes intractable for lattice field theories. The convergence of CL and the structure of the thimbles seem to be deeply interconnected, though this relationship has yet to be fully explored.

In this paper, we investigate the wrong convergence of CL by conducting a simultaneous study of thimbles and CL simulations. We observe that CL converges correctly when the system in question exhibits only one relevant compact thimble, in agreement with a previously stated conjecture [67]. Although missing a formal proof, we test this hypothesis by performing numerical simulations on a variety of toy models, which we manually regularize to control the thimble structure. To do so, we revisit a regularization (or modification) method proposed some years ago [68–70]. This additive weight regularization is based on modifying the system (and thus its thimbles) by an additive term. To correct the resulting bias from the regularization term on the expectation values, we introduce a novel method that utilizes prior knowledge of the original theory. Previously, this bias correction formed a major impediment to the original formulation of the regularization technique. We construct appropriate weight regularizations for a number of models: we study the zero-dimensional complex cosine model and multiple formulations of the (one-dimensional) Polyakov chain model for both SU(2) and SU(3). The latter serves as a test case for a lattice theory with gauge symmetry. This combined approach enables us to obtain correctly converged results for the first time for parameters and models where a direct application of the CL method fails.

This paper is structured as follows: we first discuss in detail the basics of both Lefschetz thimbles and the complex Langevin method in Section 2. We then revisit the additive weight regularization method and propose new ways to corrections for the applied regularization in Section 3. We apply these ideas then to the complex cosine model in Section 4 and the SU(2) and SU(3) Polyakov chain models in Sections 5 and 6, respectively. In Section 7, we conclude and discuss our results. The appendices include an example in Appendix A where multiple relevant compact thimbles lead to the failure of the CL method and address the question of (in-)feasibility of additive regularizations for realistic lattice field theories such as lattice QCD in Appendix B. The remaining appendices provide details of our numerical results.

## 2 The sign problem and two potential solutions

We give a short introduction to the numerical sign problem and discuss two methods that aim to circumvent or alleviate it, namely the Lefschetz thimble technique and the complex Langevin method.

We consider one-dimensional integrals of the form

$$\langle \mathcal{O} \rangle = \frac{1}{Z} \int_D dx\, \mathcal{O}(x) \exp[-S(x)]\,, \quad Z = \int_D dx\, \exp[-S(x)]\,, \tag{1}$$

where $D \subset \mathbb{R}$ is a real integration domain, $\mathcal{O} : \mathbb{R} \to \mathbb{C}$ is an observable and $S : \mathbb{R} \to \mathbb{C}$ is a complex action. The partition function $Z$ acts as a normalization constant. In this example, the complex nature of the action prohibits the interpretation of the weight $\exp[-S(x)]$ as a probability density. As a consequence, this prevents us from using Monte Carlo methods to numerically solve these types of integrals. Instead, one may split the weight into its modulus and a complex phase

$$\langle \mathcal{O} \rangle = \frac{\int_D dx\, \mathcal{O}(x) \exp[-iS_I(x)] \exp[-S_R(x)]}{\int_D dx\, \exp[-iS_I(x)] \exp[-S_R(x)]} = \frac{\langle \mathcal{O} \exp[-iS_I] \rangle_{S_R}}{\langle \exp[-iS_I] \rangle_{S_R}}\,, \tag{2}$$

where $S_R$ and $S_I$ denote the real and imaginary parts of the action, $S = S_R + iS_I$. In this form, it is, in principle, possible to use $\exp[-S_R]$ as a probability density and account for the complex phase $\exp[-iS_I]$ by reweighting. The numerical sign problem occurs whenever the imaginary part of the action becomes sizeable enough to lead to a highly oscillatory integrand within the domain $D$. In many-particle quantum systems involving fermions, this occurs when the system size or particle number is increased or the temperature is lowered. In field theories, the sign problem usually occurs at finite density or for real-time dynamics. In these cases, the relative error of the numerator on the RHS of Eq. (2), the phase factor, scales exponentially. As a result, the use of Monte Carlo methods becomes prohibitively expensive: countering the growing error of the phase factor requires an exponential increase in the required number of random samples (see e.g. [3]).

## 2.1 Lefschetz thimbles

A way to reduce the oscillatory nature of the integrand is provided by contour deformation, i.e. continuously deforming the real domain $D$ into a different integration contour $\mathcal{C}$ in the complex plane. This is possible if the action $S$ and the observable $\mathcal{O}$ can be extended to holomorphic (or at least meromorphic) functions in the complex plane. By a generalized version of Cauchy's theorem, the complex contour integral stays unchanged by such a (holomorphic) deformation of $D$ if no pole of $\mathcal{O}$ or $S$ is crossed along the way. The expectation value then reads

$$\langle \mathcal{O} \rangle = \frac{\int_{\mathcal{C}} dz\, \mathcal{O}(z) \exp[-iS_I(z)] \exp[-S_R(z)]}{\int_{\mathcal{C}} dz\, \exp[-iS_I(z)] \exp[-S_R(z)]}\,. \tag{3}$$

The sign problem can be reduced if we are able to find contours $\mathcal{C}$ along which the imaginary part $S_I(z)$ is only slowly varying. Among all possible contours, Lefschetz thimbles are particularly convenient: they are connected to the stationary points of $S$ and keep the imaginary part of the action constant. The Lefschetz thimble technique is an application of Picard-Lefschetz theory to path integrals.

The thimbles are characterized by flow equations in the complex plane. A *stable* thimble $\mathcal{J}_\sigma$ associated with the *stationary* or *critical* point $z_\sigma$, i.e. $\partial_z S(z_\sigma) = 0$, is given by the solution to the steepest-ascent equation

$$\dot{z}(t_f) = \overline{\partial_z S(z(t_f))}, \qquad t_f \in \mathbb{R}\,, \tag{4}$$

with initial condition $z(-\infty) = z_\sigma$. Here, we take the complex conjugate of $\partial_z S(z)$, which denotes the complex derivative of $S$. We classify critical points as *attractive* or *repulsive* based on the behavior of the complex derivative $-\partial_z S$ in their neighborhood: the gradient points toward attractive points and away from repulsive ones. The parameter $t_f$ is called flow time. The change of the action under this flow is given by

$$\dot{S}(z(t_f)) = \partial_z S(z(t_f)) \dot{z}(t_f) = |\partial_z S(z(t_f))|^2 > 0\,. \tag{5}$$

Since the change is real-valued, the imaginary part of $S$ must be constant along $\mathcal{J}_\sigma$. The real part of $S$ reaches its minimum in $z_\sigma$ and increases monotonically with $t_f$. Analogously, the *unstable* or anti-thimble $\overline{\mathcal{J}}_\sigma$ is defined by the steepest-descent equation

$$\dot{z}(t_f) = -\overline{\partial_z S(z(t_f))}\,, \tag{6}$$

which leads to a monotonic decrease, $\dot{S}(z(t_f)) < 0$. As an example, we plot the thimble structure of the cosine model in Fig. 1. This model will be discussed in detail in section 4.

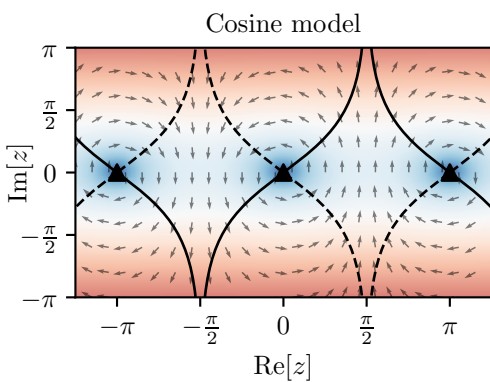

Figure 1: *Left:* Thimble structure of the cosine model with $S(z) = i\beta \cos(z)$ for real-valued couplings $\beta$. Critical points at $z_\sigma = 0, \pm\pi, \pm 2\pi, \dots$ are shown as black triangles. Extending from these critical points, the (anti-)thimbles are shown as black solid (dashed) curves. The arrows represent the direction of the Langevin drift $-\partial S$, and the background visualizes the absolute value of the drift. The cosine model exhibits $2\pi$ periodicity along the real axis.

Given these definitions, we can write the expectation value of $\mathcal{O}$ as

$$\langle \mathcal{O} \rangle = \frac{\sum_\sigma n_\sigma \exp[-iS_I(z_\sigma)] \int_{\mathcal{J}_\sigma} dz\, \mathcal{O}(z) \exp[-S_R(z)]}{\sum_\sigma n_\sigma \exp[-iS_I(z_\sigma)] \int_{\mathcal{J}_\sigma} dz\, \exp[-S_R(z)]}. \tag{7}$$

The integral is generally split into a sum over multiple contributing thimbles enumerated by the critical points $z_\sigma$ of the action. Here, $n_\sigma \in \mathbb{Z}$ counts the number of intersections of the anti-thimble with the original integration domain $D$. Thimbles with non-zero intersection numbers and their associated stationary points are referred to as *relevant*.

The Lefschetz thimble method does not remove the sign problem entirely. Each contribution of a thimble $\mathcal{J}_\sigma$ to $\langle \mathcal{O} \rangle$ is weighted by a complex phase factor $\exp[-iS_I(z_\sigma)]$. Additionally, the behavior of the observable $\mathcal{O}(z)$ along the thimble and the Jacobian factor $\dot{z}$ as in

$$\int_{\mathcal{J}_\sigma} dz f(z) = \int dt\, \dot{z}(t) f(z(t)), \tag{8}$$

lead to oscillatory integrals for each individual thimble. However, compared to the original sign problem in Eq. (1), these oscillations are typically much milder. Thus, Monte Carlo integration with probability densities $\exp[-S_R(z)] > 0$ with $z \in \mathcal{J}_\sigma$ along thimbles becomes feasible. The primary difficulty of the Lefschetz thimble method lies in determining all critical points $z_\sigma$, parameterizing all thimbles, determining the intersection numbers $n_\sigma$, and finally integrating or sampling over the thimbles. Note that in the special cases where only a single thimble $\mathcal{J}^*$ contributes, the integral simplifies to

$$\langle \mathcal{O} \rangle = \frac{\int_{\mathcal{J}^*} dz\, \mathcal{O}(z) \exp[-S_R(z)]}{\int_{\mathcal{J}^*} dz\, \exp[-S_R(z)]}. \tag{9}$$

Here, the residual complex phase $\exp[-iS_I(z^*)]$ associated with the critical point $z^*$ cancels, and the residual sign problem is further reduced.

## 2.2 Complex Langevin

Similar to the Lefschetz thimble technique, the complex Langevin method is based on continuing the integral into the complex plane. Let us first consider a real-valued action $S(x)$. Then, Eq. (1) can be approximated by the equilibrium distribution of the Langevin process

$$\frac{dx}{d\theta} = K(x(\theta)) + \eta(\theta), \qquad K(x) = -\partial_x S(x), \tag{10}$$

where $K(x)$ is the drift term and $\theta$ is an auxiliary coordinate called Langevin time. The Gaussian noise field $\eta(\theta)$ satisfies

$$\langle \eta(\theta) \rangle = 0, \tag{11}$$
$$\langle \eta(\theta)\eta(\theta') \rangle = 2\delta(\theta - \theta'). \tag{12}$$

Under quite general conditions [71], the stochastic process leads to a stationary equilibrium distribution of the form

$$P(x) = \frac{\exp[-S(x)]}{\int_D dx \, \exp[-S(x)]}, \qquad \int_D dx \, P(x) = 1, \tag{13}$$

in the limit $\theta \to \infty$. This can be shown by obtaining the stationary solution of the equivalent Fokker-Planck equation associated with the Langevin process. Thus, the process itself can be used to approximate expectation values at sufficiently late Langevin times $\theta_0$ as a stochastic integral

$$\langle \mathcal{O} \rangle = \frac{1}{\Theta} \lim_{\theta_0 \to \infty} \int_{\theta_0}^{\theta_0 + \Theta} d\theta \, \mathcal{O}(x(\theta)) = \int_D dx \, \mathcal{O}(x)P(x). \tag{14}$$

The Langevin process can be generalized to field degrees of freedom, which is known as stochastic quantization [37].

In the complex Langevin method, the process is adapted to

$$\frac{dz}{d\theta} = K(z(\theta)) + \eta(\theta), \tag{15}$$

where $K(z) = -\partial_z S(z)$ is the complex drift term. Here, we assume that $S(z)$ is holomorphic. The complex process for $z = x + iy \in \mathbb{C}$ can be understood as a process in $\mathbb{R}^2$

$$\frac{dx}{d\theta} = -\partial_x S_R(x + iy) + \eta(\theta), \tag{16}$$
$$\frac{dy}{d\theta} = -\partial_x S_I(x + iy). \tag{17}$$

As in the real case, the complex process thermalizes after some time $\theta_0$, leading to a stationary distribution $P(x, y)$. The complex Langevin method posits that the expectation values with respect to $P(x, y)$ for holomorphic observables $\mathcal{O}$ agree with the original integral over the real domain $D$

$$\langle \mathcal{O} \rangle = \frac{1}{Z} \int_D dx \, \mathcal{O}(x)\exp[-S(x)] = \int dx \int dy \, \mathcal{O}(x + iy)P(x, y), \tag{18}$$

provided that the original integral in Eq. (1) exists in the first place. In this case, the complex stochastic process can be sampled analogously to Eq. (14). The conditions under which Eq. (18) holds have been studied in recent years, which led to a better understanding of the mathematical foundation and the formulation of criteria of convergence.

Although the simplicity of the complex Langevin method is appealing for numerical simulations, there are many known examples for which the complex stochastic process fails to converge correctly. A full understanding of the sufficient requirements for this method is generally still missing and it cannot be applied blindly without involved checks to ensure correctness. Moreover, the necessary and sufficient conditions for correct convergence can usually not be checked a priori from the properties of the action $S$ and the observables $\mathcal{O}$. Instead, the criteria are formulated in terms of the stationary distribution $P(x, y)$, which is typically unknown for a general action $S(z)$.

It has been found that wrong convergence issues can occur due to the emergence of boundary terms and the order and multiplicity spectrum of the corresponding Fokker-Plank operator [59, 60]. A particularly elegant condition for correct convergence, which is necessary and sufficient, has been derived in [57]. The authors consider the magnitude of the complex drift term $u(z) = |K(z)|$ and its distribution

$$p(u; \theta) = \int dx \int dy \, \delta(u - u(x + iy))P(x, y; \theta), \tag{19}$$

where $P(x, y; \theta)$ is the probability distribution of the stochastic process at Langevin time $\theta$. If $p(u; \theta)$ falls off faster than any power law for large $u$ for all $\theta$, then the complex Langevin method converges correctly in the stationary limit. We will apply this criterion to showcase to give observable-agnostic evidence for the CL correct or wrong convergence. We note that in numerical simulations, the distribution $P(x, y; \theta)$ is, in general, not directly accessible. However, we can typically infer it from histograms of the process $z(\theta)$. In the present work, we solve the Fokker-Planck equation for the zero-dimensional models numerically to add an independent check of the criterion of correctness.

## 2.3 Connections between thimbles and complex Langevin

Similarities between the Lefschetz thimble technique and complex Langevin have been pointed out in recent years. Empirically, it has been found that complex Langevin may converge correctly if the equilibrium process stays close to the relevant thimbles of the action [64]. At first sight, this is plausible because the critical points of the action are central to both methods. In the case of the thimble technique, the critical points are defined as the start points of the steepest-ascent/descent flow. Within complex Langevin, the critical points lead to a vanishing of the drift term. For attractive critical points, the equilibrium process typically stays close to these points. It has also been suggested that the failure of complex Langevin may be attributed to thimbles escaping to imaginary infinity at finite values along the real line [65], in contrast to *compact* thimbles that stay close to the real domain. This is indeed the case in the cosine model for any complex value of the coupling, see Fig. 1. In other systems such as the Polyakov chain, the thimble structure changes depending on the parameters of the system, as we will discuss in Sections 5 and 6.

A breakup of the thimble structure is typically related to multiple relevant thimbles and, accordingly, multiple relevant stationary points. Further analytical evidence for this has been found in [66], where a semi-classical analysis of both the complex Langevin and the thimble method has revealed that the complex stochastic process may fail to converge correctly if the action for the system exhibits multiple dominant critical points. In these cases, failure of convergence appears to be due to the complex process incorrectly sampling the individual complex phases of the contributing thimbles. The authors of [66] suggested a reweighting method to correct for the relative phases of the critical points and found, at least in simple models, that correct convergence is indeed restored. Unfortunately, the proposed reweighting method relies on prior knowledge of the critical points. Except for simple toy models, determining all

critical points of a lattice field model is numerically infeasible. The numerical and analytical evidence so far strongly suggests that complex Langevin generically fails to converge if the system exhibits a multi-thimble structure. A formal proof of this is, however, still missing.

## 3 Weight regularization

A potential solution to overcome these convergence issues is to introduce an artificial regularization by adjusting the thimble structure, enabling the complex Langevin method to achieve correct convergence. This approach follows the ideas presented in a series of works [68–70], which propose regularizing the weight function, $\rho(z) = \exp[-S(z)]$, through an additive modification. Specifically, we consider regularizations

$$\tilde{\rho}(z) = \rho(z) + R(z), \qquad R(z) = r\, G(z) + R_0\,,\tag{20}$$

where $G : \mathbb{C} \to \mathbb{C}$ is a holomorphic function, $r \in \mathbb{C}$ is a parameter controlling the strength of the regularization and $R_0 \in \mathbb{C}$ is a constant. The main idea is that the regularization term $R(z)$ is designed in such a way that the regularized weight $\tilde{\rho}(z)$ exhibits exactly one compact relevant thimble, thus enabling correct convergence within complex Langevin.

For the regularized system, expectation values of observables $\mathcal{O}$ are given by

$$\langle \mathcal{O} \rangle_{\tilde{\rho}} = \frac{\int_D dx\, \mathcal{O}(x)[\rho(x) + R(x)]}{\int_D dx\, [\rho(x) + R(x)]} = \frac{\int_D dx\, \mathcal{O}(x) \exp[-\tilde{S}(x)]}{\int_D dx\, \exp[-\tilde{S}(x)]}\,,\tag{21}$$

where $\tilde{S}(z)$ may be interpreted as a regularized action given by

$$\tilde{S}(z) = -\ln[\rho(z) + R(z)] = S(z) - \ln[1 + R(z)e^{S(z)}]\,.\tag{22}$$

Accordingly, expectation values computed for the regularized system, Eq. (21), generally differ from the original system. We interpret the difference between the original and the regularized expectation values as a bias term,

$$\langle \mathcal{O} \rangle_\rho = \langle \mathcal{O} \rangle_{\tilde{\rho}} + \text{Bias}_{\mathcal{O}}[R]\,.\tag{23}$$

There are two special cases to consider: Clearly, for $r = 0$ and $R_0 = 0$, the original action is recovered and the bias vanishes. On the other hand, in the limit $|r| \to \infty$, the regularization term $R$ dominates over the original weight $\rho$. In this case, the expectation values computed within the regularized system correspond to a system with weight $R$.

### 3.1 Bias correction

In order to connect the modified expectation values to the original ones, we rewrite Eq. (21) in the following way:

$$\langle \mathcal{O} \rangle_{\tilde{\rho}} = \frac{Z_\rho \langle \mathcal{O} \rangle_\rho + Z_R \langle \mathcal{O} \rangle_R}{Z_\rho + Z_R} = \frac{\langle \mathcal{O} \rangle_\rho + Q \langle \mathcal{O} \rangle_R}{1 + Q}\,,\tag{24}$$

where

$$\langle \mathcal{O} \rangle_R = \frac{1}{Z_R} \int_D dx\, \mathcal{O}(x) R(x)\,, \qquad Z_R = \int_D dx\, R(x)\,, \qquad Z_\rho = \int_D dx\, \rho(x)\,,\tag{25}$$

and the coefficient $Q$ is given by the ratio of the partition functions, $Q = Z_R/Z_\rho$. By rearranging Eq. (24), the original expectation value $\langle \mathcal{O} \rangle_\rho$ can be written as

$$
\begin{aligned}
\langle \mathcal{O} \rangle_\rho &= \langle \mathcal{O} \rangle_{\tilde{\rho}} + \text{Bias}_{\mathcal{O}}[R] = \langle \mathcal{O} \rangle_{\tilde{\rho}} + (\langle \mathcal{O} \rangle_{\tilde{\rho}} - \langle \mathcal{O} \rangle_R) Q \\
&\equiv \langle \mathcal{O} \rangle_{\tilde{\rho}}^{\text{bias-corr.}}.
\end{aligned}
\tag{26}
$$

Equation (26) shows that expectation values of observables $\mathcal{O}$ of the original system can be expressed in terms of expectation values of the regularized systems and an observable-independent ratio $Q$. Provided that the Langevin processes converge correctly for the regularized weight $\tilde{\rho}$ and the regularization term $R$, the only remaining unknown is the ratio of the partition functions. Clearly, $Z_\rho$ can be difficult to estimate because it is affected by the numerical sign problem.[1] We find that there are, nonetheless, methods that allow us to extract the ratio $Q$ indirectly. Once $Q$ is computed to sufficient accuracy, the bias correction formula (26) can be applied to any observable. Here, we examine two methods of determining $Q$ self-consistently.

**Method 1: $r$-dependence of expectation values.** The regularized weight function $\tilde{\rho}$ is an affine function in the regularization parameter $r$. This property can be used to extract the ratio $Q$. In particular, for $R_0 = 0$ we have

$$
Q = Z_R/Z_\rho = r Z_G/Z_\rho = r \tilde{Q},
\tag{27}
$$

where $Z_G = \int_D dx\, G(x)$ and $\tilde{Q} = Z_G/Z_\rho$. Assuming we have two regularized, correctly converging systems with different regularization parameters $r_1$ and $r_2$, then we can explicitly solve Eq. (26) for $\tilde{Q}$. Using $\langle \mathcal{O} \rangle_R = \langle \mathcal{O} \rangle_G$, we find

$$
\tilde{Q} = \frac{\langle \mathcal{O} \rangle_{r_2} - \langle \mathcal{O} \rangle_{r_1}}{r_1 \langle \mathcal{O} \rangle_{r_1} - r_2 \langle \mathcal{O} \rangle_{r_2} - (r_1 - r_2)\langle \mathcal{O} \rangle_G},
\tag{28}
$$

where $\langle \mathcal{O} \rangle_r$ refers to the regularized system with parameter $r$. The above equation makes no explicit reference to $Z_\rho$, or even $Z_R = r Z_G$. Thus, all required quantities can be computed using complex Langevin.

We can extend this idea beyond just two measurements: Consider a set of parameters $\{r_1, r_2, \ldots, r_N\}$ and associated expectation values $\{\langle \mathcal{O} \rangle_{r_1}, \langle \mathcal{O} \rangle_{r_2}, \ldots, \langle \mathcal{O} \rangle_{r_N}\}$. We can perform a fit to these measured data using the right-hand side of Eq. (26). Thus, we can extract $\langle \mathcal{O} \rangle_\rho$, $\langle \mathcal{O} \rangle_G$ and the ratio $\tilde{Q}$, simply by running multiple simulations at various regularization parameters $r_i$. Performing a statistical fit also allows us to determine the uncertainty of the ratio $Q$. We note that this method is very similar to the "multi-modification method" proposed in [70]. For $R_0 \neq 0$, the same methods can be used, but we require at least three different values of the regularization strength $r$.

In practice, we find that determining $Q$ in this way can be statistically very difficult. Depending on the original action $S$ and the severity of the sign problem, the Langevin process may require rather strong regularizations for correct convergence. In these situations, the dependence of $\langle \mathcal{O} \rangle_r$ on $r$ might be very weak and, further, the observables computed within the regularized system may essentially correspond to $\langle \mathcal{O} \rangle_R$. In this case, extraction via Eq. (28) becomes unreliable and leads to a large statistical error.

---

[1]We remark that the computation of $Z_\rho$ represented one of the key limitations in [68], where weight regularizations were proposed to improve the convergence of CL.

**Method 2: Dyson-Schwinger equations (DSE).** The ratio $Q$ can also be extracted by utilizing its observable independence. For an observable $\mathcal{O}^*$ whose expectation value vanishes for the original system, we can solve Eq. (26) for $Q$ to obtain

$$Q = \frac{\langle \mathcal{O}^* \rangle_{\tilde{\rho}}}{\langle \mathcal{O}^* \rangle_R - \langle \mathcal{O}^* \rangle_{\tilde{\rho}}}, \tag{29}$$

where we assume that the expectation value $\mathcal{O}^*$ in the regularized system does not vanish. Such observables can be constructed systematically using the Dyson-Schwinger equations associated with the action $S$. Given some holomorphic function $\mathcal{O}(z)$, the Dyson-Schwinger equations read

$$-\langle \mathcal{O} \partial_x S \rangle_\rho + \langle \partial_x \mathcal{O} \rangle_\rho = \frac{1}{Z} \int_D dx \, \partial_x(\rho(x) O(x)) = 0, \tag{30}$$

as the total derivative vanishes. This implies that

$$\mathcal{O}^* \equiv -\mathcal{O} \partial_z S + \partial_z \mathcal{O}, \tag{31}$$

has the desired property. Analogously to the first method, we may consider multiple observables $\mathcal{O}^*$ and multiple parameters $r$ to improve and estimate the accuracy of $Q$, but, in principle, only a single simulation at a specific parameter $r$ is required. Our computational experiments indicate that this method is more reliable than the previous approach. As we will demonstrate, a single set of Dyson-Schwinger equations along with a single parameter, $r$, suffices to extract $Q$ with reasonable numerical accuracy. It is important to note, however, that as $|r| \to \infty$, the denominator of Eq. (29) approaches zero, potentially making the computation intractable. To counter this, we carefully tune $r$ to ensure that we obtain a single compact relevant thimble while avoiding an excessive increase in its magnitude. This tuning approach maintains the statistical stability of $Q$ for all tested models.

## 3.2 Designing appropriate regularizations

The goal of regularizing the weight $\rho$ with a regularization term $R$ is to obtain a modified weight function with improved convergence properties with regard to CL simulations. Our numerical studies suggest that the complex Langevin process tends to correctly converge if there exists a single relevant, attractive critical point and if the relevant thimble extending from the relevant critical point is compact. In general, choosing an appropriate regularization is non-trivial, but there are a few principles that can guide the design process.

First of all, it is useful to consider *real* regularization functions $G(z)$, which are positive definite within the original domain $z \in D \subset \mathbb{R}$. In the limit $|r| \to \infty$, the thimble structure of $\tilde{\rho} = \rho + R_0 + r G \approx r G$ is entirely determined by the regularization term. Since $G$ exhibits no sign problem, the thimble of $\tilde{\rho}$ coincides with the real domain $D$. Clearly, complex Langevin works in this special case because it reduces to real Langevin. Thus, depending on the specifics of the regularization term, there may exist a finite critical parameter $r_c$ beyond which there is only a single relevant critical point and thimble.

Secondly, we can use the constant $R_0$ to introduce zeros in the regularized weight $\tilde{\rho}$ to improve the thimble structure. Generally, these singularities in the corresponding effective action are troublesome for complex Langevin, as they introduce singular points in the drift. Unless the stationary distribution of the complex Langevin process decays sufficiently fast in the neighborhood of the singularities of the drift, the criterion of correctness may be violated. On the other hand, zeros in $\tilde{\rho}$ can be used to "capture" thimbles that would otherwise escape to imaginary infinity. Extending outward from the critical points, thimbles generically have to

either asymptotically approach infinity or connect to a zero of the weight function.[2] By introducing attractive singularities, these thimbles may be "tamed" such that the thimble structure is connected. Moreover, if the constant $R_0$ and the function $G$ are chosen such that these new singularities happen to be at the boundaries of the real domain $D$, then, at least intuitively, one could expect the singular drift problem to be not severe. For example, the gauge-invariant formulation of the Polyakov chain discussed in sections 5 and 6 naturally exhibits singularities that stem from the Haar measure. At least for some parameter choices, these do not appear to invalidate the criterion of correctness. In fact, all regularizations considered in this paper introduce artificial zeroes in the weight function. Following this guiding principle, we find that the constant $R_0$ is generally complex and thus introduces a sign problem into the regularization term $R$. Therefore, one has to check additionally if the complex Langevin process for the weight $R$ is stable and converges correctly.

In summary, we construct the weight regularizations using the following steps:

1. We determine zeros of the original weight function.

2. We consider integrable and positive definite functions $R$ on the real integration domain such that $R$ and $\rho + R$ have only zeros at the boundary of this domain. In the case of point symmetry, as in the models we consider, another zero lies at the origin.

3. The drift associated with the complex continued function $R$ should point towards the real line everywhere. For strong regularization with $|r| \to \infty$, this effectively squeezes the stochastic process towards the real line.

4. If all the singularities in the neighborhood of the real line are at the boundary for a sufficiently large $|r|$, there should now only be one compact thimble that connects those singularities. However, situations may occur where the desired single thimble structure is asymptotically reached for $|r| \to \infty$, while multiple relevant compact thimbles are present for finite $|r|$ (see Appendix A). Therefore, the thimble structure of the weight regularized theory should be checked explicitly.

## 4 Complex cosine model

Our first application of the regularization method is the complex cosine model given by the action

$$S(x) = i\beta \cos(x), \quad x \in D = [-\pi, +\pi], \quad \beta \in \mathbb{R}. \tag{32}$$

The thimble structure is shown in Fig. 1. The model exhibits critical points on the real line at $x \in \{0, \pm\pi, \pm 2\pi, \dots\}$. The thimbles extending from the critical points are non-compact, i.e. they extend to imaginary infinity at $x \in \{\pm\pi/2, \pm 3\pi/2, \dots\}$. Since the real-valued coupling constant $\beta$ only appears as an overall factor in the action, the thimble structure does not depend on the specific value of the coupling. The cosine model is particularly interesting in the context of complex Langevin because it is one of the simplest examples in which the stochastic process fails to converge correctly, irrespective of the chosen coupling constant $\beta$. Moreover, it is exceptional because an analytical solution for the stationary distribution $P(x, y)$ has been found in [67]

$$P(x, y) = \frac{1}{4\pi} \frac{1}{\cosh^2(y)}. \tag{33}$$

---

[2]Unless they connect to another critical point, which is termed the Stokes phenomenon.

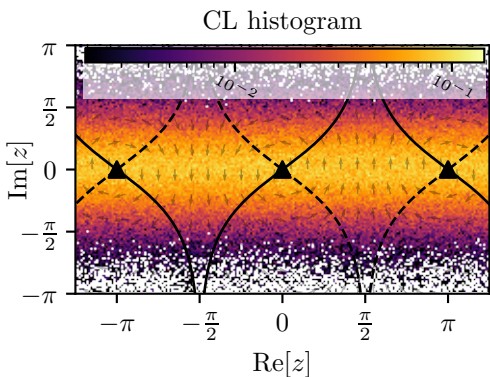

Figure 2: Histogram on a logarithmic scale of a complex Langevin process for the complex cosine model at $\beta = 0.5$ without regularization. Arrows show the normalized CL drift, while solid and dashed curves represent thimbles and anti-thimbles of critical points (black triangles).

A likely reason for the failure of complex Langevin in this model is the non-compact thimble structure. Our goal is to design a regularization that tames these thimbles in the sense that the regularized model exhibits a compact one-thimble structure. In Fig. 2 we show the histogram of a complex Langevin process alongside the thimble structure. The histogram reproduces the analytical solution (33) of the corresponding Fokker-Planck equation. We emphasize that the stationary points of this model are center points and are, thus, neither attractive nor repulsive. In consequence, the stationary distribution is independent of the real part $x$ as both stationary points have equal weight. The wrong convergence of this model was showcased based on boundary terms and expectation values of observables in [58].

To cure this model of wrong convergence, we consider the regularization term

$$R(z) = r(z^2 - \pi^2) - e^{i\beta}, \quad z = x + iy, \qquad x \in D, y \in \mathbb{R}. \tag{34}$$

The modification is chosen such that for $|r| > 0$, $\tilde{\rho} = \rho + R$ exhibits an attractive critical point at $z = 0$ and zeros at $z = \pm\pi$, i.e. $\tilde{\rho}(\pm\pi) = 0$. The subtraction of the constant $e^{i\beta}$ was introduced to impose the latter. Thus, the Lefschetz thimbles starting at $z = 0$ must eventually end in the singular points at the boundary of the real domain $D$. At large enough values of $r$, the modification pulls the thimbles close enough to the real domain $D$ such that they can not escape to infinity. We stress that the location of the singularities on the boundary of the original domain of integration is essential: if the singular points are not located on the boundary of $D$, it is not guaranteed that $D$ can be deformed continuously to a single and compact steepest-descent curve for sufficiently large $r$. This might introduce a non-vanishing probability density at the boundary. Moreover, to ensure periodicity along the real line, we restrict the regularization term periodically

$$R(x + iy + 2\pi) = R(x + iy), \qquad \forall (x, y) \in \mathbb{R}^2. \tag{35}$$

The thimbles of the regularized model are shown in the left panel of Fig. 3. Correspondingly, the right panel of Fig. 3 shows the histogram of the CL process and the obtained one-thimble structure for $\beta = 0.5$ and $r = 0.5$. We observe that the histogram is located around the relevant thimble but does not follow its precise form in the complex plane. We also find that if the relevant thimble winds closer to the real domain, we obtain a sharper histogram in the imaginary direction leading to better-conditioned convergence properties.

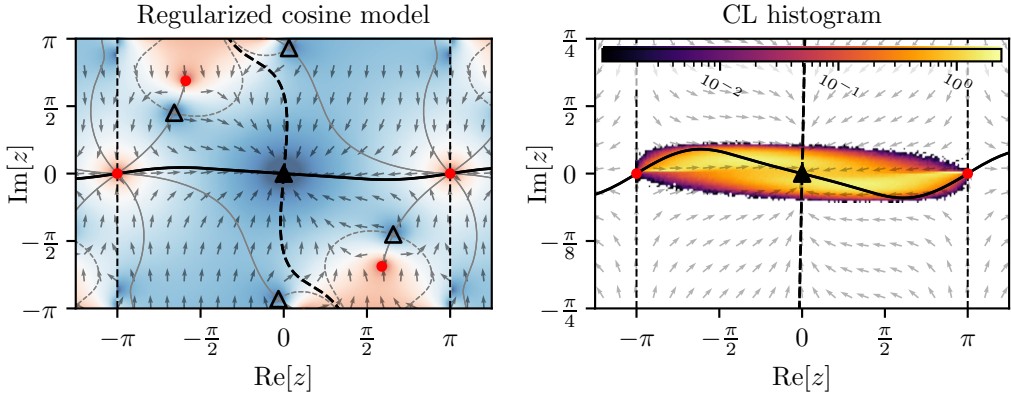

Figure 3: *Left panel:* Thimble structure of the regularized cosine model with $\beta = 0.5$ and $r = 0.5$. As in Fig. 1, the black triangles are critical points. Solid and dashed curves represent thimbles and anti-thimbles. The modified model exhibits multiple critical points, but only the point at the center contributes. The non-contributing points are shown as unfilled triangles, and their associated thimbles are shown in gray. The contributing thimble extending from $z = 0$ connects to two singularities placed at $\pm\pi$ (red dots). Additional singularities are also shown but are not connected to any contributing thimbles. The vertical dashed lines at $Re[z] = \pm\pi$ indicate the periodic boundaries. *Right panel:* The CL histogram for the regularized cosine model for the same parameters appears to be compact in the imaginary direction.

Imposing periodicity in Eq. (34) leads to a non-holomorphic effective action. At $x = \pm\pi$ and $y \in \mathbb{R}$, the function $R$ and thus the weight $\tilde{\rho}$ are not differentiable. It appears that the solution seems to be unaffected by these results. This may be due to several aspects: the complexified model may not be seen as an analytical continuation to the whole complex plane but rather to the open box $(-\pi, \pi) + i\mathbb{R}$ without the isolated singularities of the effective action. We observe that the singularities and the boundary segments $\pm\pi \times (-\delta, \delta)$ are repulsive, with $\delta$ growing with $r$, so the probability weight of the stationary solution along the boundary vanishes, making it irrelevant to the results. We note that a model similar to the complex cosine model has already been studied in [68] using the weight regularization method with $R(z) \propto (z^2 - \pi^2)e^{-S(z)}$. The authors impose periodicity by hand as in Eq. (35) and still recover the correctly covering complex Langevin distribution. This confirms that periodic continuation is not problematic in certain scenarios.

In the following, we provide our simulation details and compare the extracted results with analytical values from a direct solution to the model. We will then check the criterion of correctness for the cosine model directly, demonstrating the exponential decay of density for the drift magnitude for the regularized model.

## 4.1 Bias correction

We test the regularization approach with coupling $\beta = 0.5$ and regularization strength $r = 0.5$, simulating $2^{16} \approx 6.5 \cdot 10^4$ independent CL processes.[3] Each process is initialized randomly within $(-\pi, \pi)$ using a uniform distribution.[4] To obtain stationary solutions, the CL processes are simulated until $\theta = 100$, beyond which no further changes in the probability density were

---

[3]We prioritize the generation of independent processes over the length of each process to efficiently parallelize our simulation on a GPU.

[4]We also tested initialization over a complex box $(-\pi, \pi) + i(-3\pi, 3\pi)$ that spans the action's singularities. The results were unaffected, matching those from initialization along the real line.

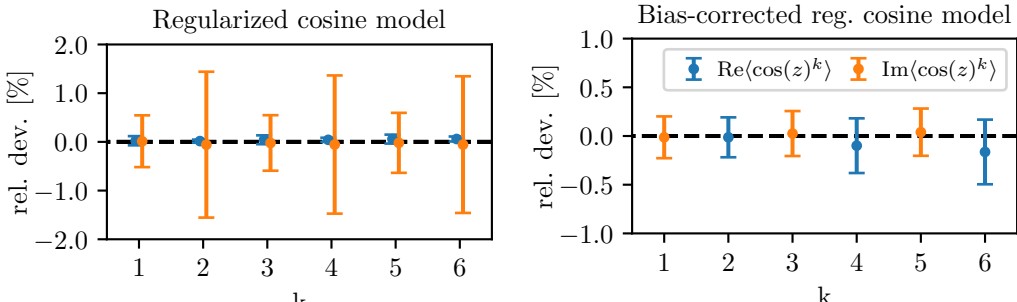

Figure 4: Relative deviation of expectation values of $\langle\cos(z)^k\rangle$ $(k = 1,\ldots,6)$ from the respective analytical values for the regularized CL model with $\tilde{\rho}$ (*left*) and for bias-corrected values via Eq. (26) (*right*). The left panel shows that weight regularization can lead to correct results with respect to the regularized weight function while the right panel demonstrates that the bias-corrected values accurately agree with the original theory. Only non-vanishing real/imaginary parts of orders $k$ are shown. The corresponding exact values and our CL results are listed in Tables 2 and 3.

observed (thermalization). The evolution uses a step size $\epsilon = 10^{-5}$, adjusted adaptively as $\tilde{\epsilon}(\theta) = \min(\epsilon, \alpha/|K(z(\theta))|)$ with $\alpha = 10^{-2}$, allowing a maximum drift magnitude of $|K| = 10^3$. This adaptive step size prevents numerical instabilities that typically occur close to singularities or when the process drifts too far into the complex plane. After the thermalization phase, samples of the CL processes are taken every $\Delta\theta = 1$, collecting 100 samples in total per process. This setup simplifies statistical analysis, as each process provides an independent sample average for expectation values.

To extract expectation values for the original model, we follow Eq. (26) and calculate the ratio $Q$ using the Dyson-Schwinger equation with $\mathcal{O} = \sin(x)$. This yields the observable

$$\mathcal{O}^* = i\beta\sin^2(x) + \cos(x), \tag{36}$$

that has a vanishing expectation value with respect to the original model $\rho$. To compute $Q$, we evaluate $\langle\mathcal{O}^*\rangle_{\tilde{\rho}}$ and $\langle\mathcal{O}^*\rangle_R$, then substitute these into Eq. (29). Note that as $r \to \infty$, the sign problem lessens; however, this also diminishes the information from the original model, causing the denominator in Eq. (29) to approach zero. Therefore, we choose $r$ to be strong enough to enforce a compact one-thimble structure and correct convergence, but small enough to preserve meaningful information for calculating $Q$.

In Fig. 4, we compare our numerical CL results with analytical values that can be obtained by direct integration, see the tables in Appendix C. The left panel of Fig. 4 shows the relative deviation of $\langle\cos(x)^k\rangle$ for $k = 1,\ldots,6$ with respect to the regularized weight $\tilde{\rho}$. The right panel shows the deviation with respect to the bias-corrected expectation values using Eq. (26). For each $k$, we show the non-vanishing part: odd orders for the imaginary part and even orders for the real part, as the others vanish identically and prevent the calculation of relative deviations. We find that the CL method correctly reproduces the expectation values of the regularized model and that the results from the suggested bias correction accurately agree with the original theory.

We emphasize that applying CL directly to the original cosine model without a regularization yields diverging results for $k > 2$ and wrong results for $k = 1, 2$ [58]. Our results indicate that this regularization and correction approach enables the CL method to be applied to models previously deemed unsolvable with CL, accurately yielding correct expectation values even for high-order observables.

## 4.2 Direct check of the criterion of correctness

Incorrect convergence in complex Langevin can arise without any numerical pathologies in the observables that would typically indicate distorted results. The formulation of criteria of correctness allows us to diagnose the convergence of complex Langevin based on observable-dependent boundary terms [58] or the decay of the density for the drift magnitude [57]. The latter criterion is agnostic of observables and states that the stationary solution of the complex Langevin equation characterizes the desired weight function $\rho \propto \exp(-S)$ correctly if the density for the drift magnitude $p(u, \theta)$ stated in Eq. (19) decays at least exponentially fast. We can, therefore, determine numerically whether the criterion of correctness is satisfied for a given (regularized) weight in an observable-independent manner.

As mentioned earlier, the cosine model is a special case, because its stationary solution, $P(x, y) = 1/(4\pi \cosh^2(y))$, is independent of both $x$ and the coupling. This $x$-independence suggests that complex Langevin may produce incorrect results, as the path integral assigns different weights across the integration domain. In Figure 5, we show that the correctness criterion is not met: We plot $p(u)$ without a regularization term and observe that the drift density decays as a power law, even though $P(x, y)$ decays exponentially in the imaginary direction. In contrast, with a regularization $r = 0.5$, the drift density exhibits exponential decay, which indicates that the criterion of correct convergence is satisfied and the CL method produces correct results. Although no explicit analytical link has yet been established, in all tested models, we observe that a compact one-thimble structure aligns with meeting the correctness criterion.

We note that for the drift magnitude of the unregularized cosine model, we sample directly from the real probability density $P$ to construct the histogram. For more complex models, the histogram can be obtained by numerically solving either the complex Langevin equation or its associated Fokker-Planck equation. In the case of one-variable models, we solve the Fokker-Planck equation directly using an implicit scheme[5] as an independent check of our simulation. This approach helps to avoid potential issues like thermalization effects, problems with ergodicity, or artifacts associated with the numerical scheme used to solve the stochastic differential equations.

## 5 SU(2) Polyakov chain model

The second model considered in this study is the one-dimensional SU(2) Polyakov chain model, which serves as a toy model for lattice gauge theories exhibiting a sign problem. Notably, without the application of stabilization techniques [73], the complex Langevin (CL) approach has previously been observed to fail. The action reads

$$S[U] = -\beta \operatorname{Tr}[P], \tag{37}$$

with $\beta \in \mathbb{C}$ and where the Polyakov chain $P$ is given by the product of $N_{\text{chain}}$ links

$$P = U_1 U_2 \cdots U_{N_{\text{chain}}}, \quad U_i \in \mathrm{SU}(2), \quad i \in \{1, 2, \dots, N_{\text{chain}}\}. \tag{38}$$

Under gauge transformations $U_i \to \Omega_i U_i \Omega_{i+1}^\dagger$ with $\Omega_i \in \mathrm{SU}(2)$, the action remains invariant. Therefore, expectation values of gauge invariant observables $\mathcal{O}[P]$ are given by

$$\langle \mathcal{O} \rangle_\rho = \frac{1}{Z_\rho} \int \prod_i^{N_{\text{chain}}} DU_i \, \mathcal{O}[P] \exp\{-S[U]\}, \tag{39}$$

where $DU_i$ is the SU(2) Haar measure.

---

[5]The implicit solver ensures that the probability flux at the boundaries vanishes to conserve the integral over probability density and enforces the positivity of the solution. It is based on the Chang-Cooper scheme [72] with exponential upwind to prevent instabilities. Details on this scheme will be published in future work.

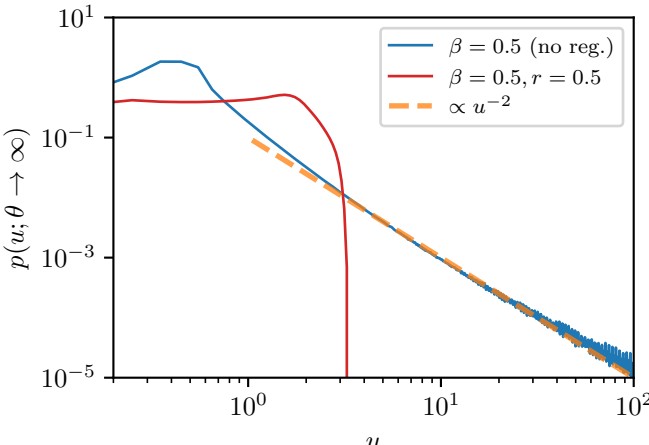

Figure 5: The density $p(u)$ of the drift magnitude $u$ from Eq. (19) is shown for both the original (blue) and regularized (red) cosine models at $\beta = 0.5$. The original model exhibits a power law decay (indicated by the orange dashed line), which violates the convergence criterion, while the regularized model demonstrates exponential decay, indicating correct convergence of the complex Langevin method.

The degrees of freedom in the Polyakov chain model are highly redundant due to gauge symmetry since the gauge freedom of the action allows reducing the Polyakov chain to a one-link model. Moreover, the Haar-measure for SU(2) can be further simplified by parametrizing the matrix elements using the 3-sphere $S^3$ and integrating out all angles except for one [74]. This leads to the identification

$$\mathrm{Tr}[P] \to 2\cos(\phi), \tag{40}$$

$$DU \exp\{-S[U]\} \to d\phi \sin^2(\phi)\exp\{2\beta\cos(\phi)\} =: d\phi J(\phi)\rho(\phi), \tag{41}$$

where $\phi$ is integrated over the interval $[-\pi, \pi]$ and $J(\phi) = \sin^2(\phi)$ denotes the Jacobian term for the chosen coordinates. We refer to this model as the *reduced SU(2) Polyakov chain model* with the weight function $\rho$. This identification allows the direct numerical computation of observables. Therefore, it represents a well-suited model for diagnosing CL's wrong convergence and provides insights into the design of weight regularizations for gauge theories.

We now have two ways of performing the CL simulations. For the reduced model, the numerical simulations are performed in full analogy to the cosine model (see Sec. 4.1 for details). Only the drift needs to be adapted to respect the effective action of the reduced Polyakov chain model. For the SU(2) Polyakov chain model, i.e. the link formulation, we employ $N_{\mathrm{chain}} = 64$ and an adaptive step $\tilde{\epsilon}(\theta) = \min(\epsilon, \alpha/\|K[U(\theta)]\|)$ with $\alpha = 10^{-2}$ and a maximal Langevin time step of $\epsilon = 10^{-4}$. Here, $\|\cdot\|$ denotes the Frobenius norm. We simulate $10^4$ independent trajectories that are initialized by identity matrices and measure observables after a thermalization time of $\theta_{\mathrm{therm}} = 100$ after every in intervals of $\Delta\theta = 0.1$ until $\theta = 1000$. After each CL step, we perform 10 adaptive gauge-cooling steps [73] with a gradient step size of $\alpha_{\mathrm{GC}} = 0.1$ to minimize the unitarity norm $F[U] = \sum_{i=1}^{N_{\mathrm{chain}}} \mathrm{Tr}[U^\dagger U - \mathbb{1}]$. Notably, this does not affect the unitary gauge freedom and does not reduce the Polyakov chain to a one-link model.

The reduced Haar measure in Eq. (41) introduces a Jacobian that enters as a logarithmic term in the (effective) action

$$S_{\mathrm{eff}}(\phi) = -2\beta\cos(\phi) - \ln\left[\sin^2(\phi)\right], \tag{42}$$

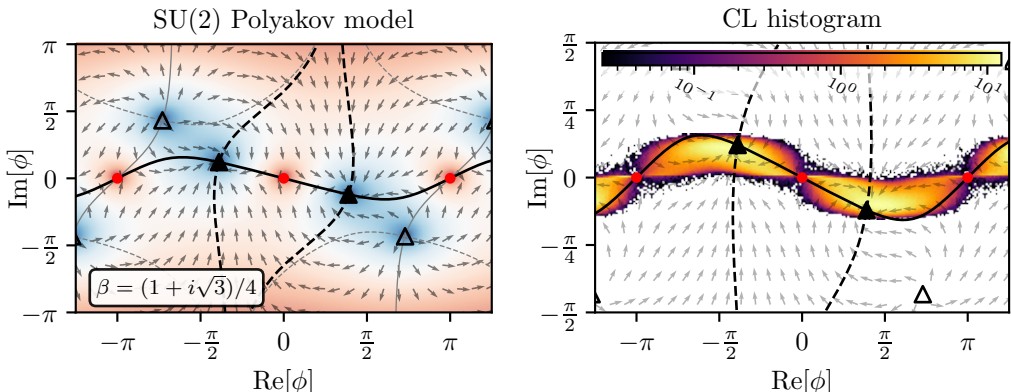

Figure 6: Thimble structure of the reduced SU(2) Polyakov chain model for the coupling $\beta = (1 + i\sqrt{3})/4$ *(left)* and the corresponding histogram from our CL simulations *(right)*. We use the same conventions as in Figs. 1 and 3. Up to point symmetry around $\phi = 0$, only one relevant thimble exists, around which the CL process samples within a compact domain.

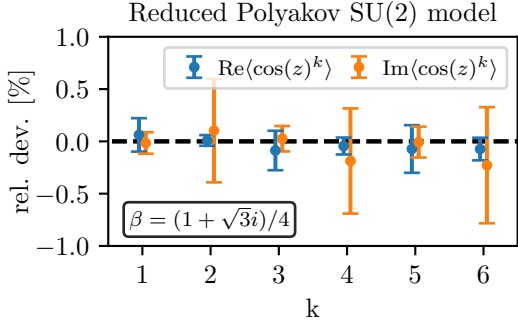

Figure 7: Relative deviation of expectation values of $\langle \cos(z)^k \rangle$ ($k = 1, \ldots, 6$) from the corresponding analytical values for the reduced SU(2) Polyakov chain model with $\beta = (1 + \sqrt{3}i)/4$. For this coupling, the CL process is seen to converge correctly. The values are listed in Table 4.

which leads to singular points at $\phi_s \in \{0, \pm\pi, \pm 2\pi, \ldots\}$. Unlike the cosine model, the thimble structure depends on the value of the complex coupling constant $\beta$.

In the left panel of Fig. 6, we show the thimble structure of the reduced formulation for the coupling $\beta = (1 + i\sqrt{3})/4$, which was also studied in [65, 73]. The CL process converges correctly in this setting. This is indicated by the compact histogram presented in the right panel of Fig. 6 that roughly follows the thimble structure. We also confirm the convergence with the relative deviation of expectation values of $O_k(\phi) = \cos(\phi)^k$ from the exact values in Fig. 7. The visible consistency of the values confirms the correct convergence of the CL process.

When the magnitude of the coupling increases by a factor of two to $\beta = (1 + i\sqrt{3})/2$, the model undergoes a Stokes phenomenon roughly at $\beta \approx (1 + i\sqrt{3})/2.7072$, which leads to four relevant stationary points instead of two as before.[6] This is shown in the top left panel of Fig. 8. All four corresponding relevant thimbles (black solid lines) are not compact since they spread out in the imaginary direction. Correspondingly, the histogram of the thermalized

---

[6]At the critical coupling of the Stokes phenomenon, both stationary points in the left and right half-planes become relevant and are connected by a Lefschetz thimble.

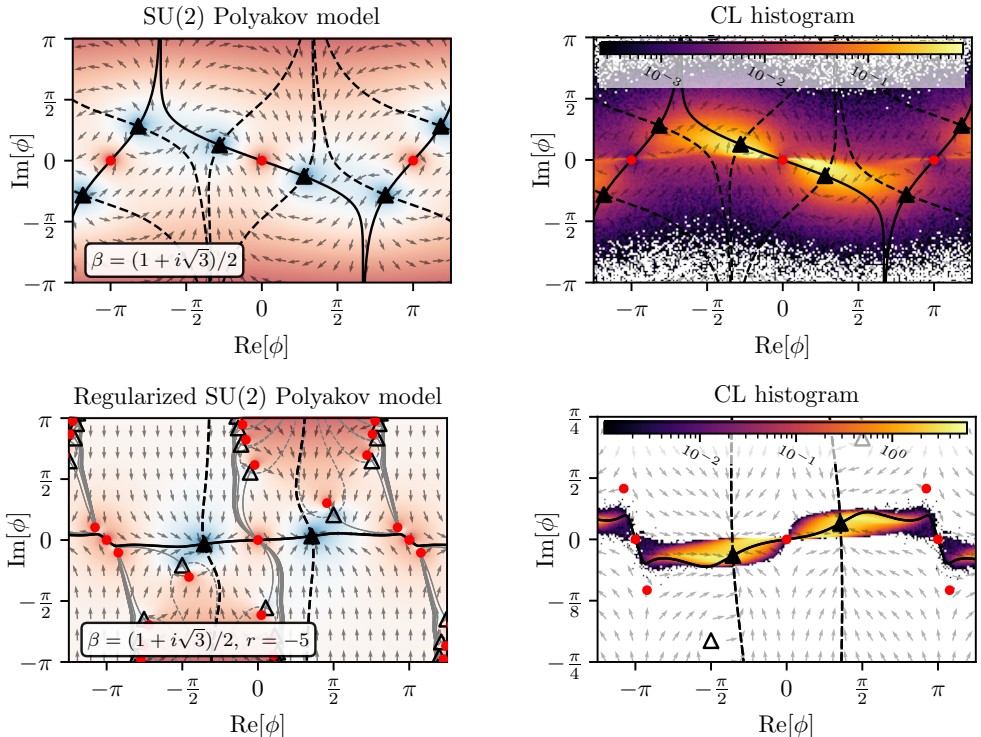

Figure 8: Thimble structures *(left)* and the corresponding CL histograms *(right)* of the reduced SU(2) Polyakov chain model for the coupling $\beta = (1+i\sqrt{3})/2$. The same conventions as in Figs. 1 and 3 are used. The relevant thimbles of the unregularized formulation are non-compact leading to a slow decay of the CL histogram in the imaginary directions *(top)*. In contrast, weight regularization using Eq. (43) with $r = -5$ yields compact relevant thimbles and a compact CL histogram *(bottom)*, indicating correct convergence.

CL process shown in the top right panel of Fig. 8 follows these thimbles and decays only slowly towards imaginary infinity. This slow decay breaks the criterion of correctness and hence yields incorrect results as can be seen in Table 5. In the following, we introduce an appropriate weight regularization that cures the wrong convergence and allows the extraction of correct expectation values using the CL approach.

## 5.1 Weight regularization for the reduced model

As in the complex cosine model, we introduce an additive regularization term to the weight function for the reduced SU(2) Polyakov chain model

$$
\begin{aligned}
\tilde{\rho}(\phi) &= \rho(\phi) + R(\phi) \\
&= e^{2\beta \cos(\phi)} + r(\cos(\phi) + 1) \, .
\end{aligned}
\tag{43}
$$

The periodic regularization term $R$ vanishes at the integration boundaries $\phi = \pm\pi$, where the reduced Haar measure causes singularities in the effective action. To prevent connections between multiple contributing thimbles, $R$ is designed to have no non-real roots. The stochastic process follows the effective action of the regularized weight function including the Jacobian $J(\phi) = \sin^2(\phi)$ that yields the CL drift:

$$
K(\phi) = \partial_\phi \ln[J(\phi)(\rho(\phi) + R(\phi))] = \frac{J'(\phi)}{J(\phi)} + \frac{\rho'(\phi) + R'(\phi)}{\rho(\phi) + R(\phi)} \, .
\tag{44}
$$

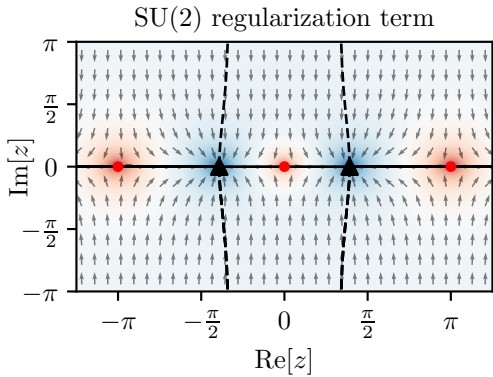

Figure 9: Thimble structure of the regularization term $R(\phi)$ of Eq. (43), including the Jacobian $J(\phi)$, for the reduced SU(2) Polyakov model in the complex plane. The drift $K = \partial_\phi \ln[J(\phi)R(\phi)]$ pulls the CL process toward the real line. The relevant thimbles (solid curves) coincide with the real line.

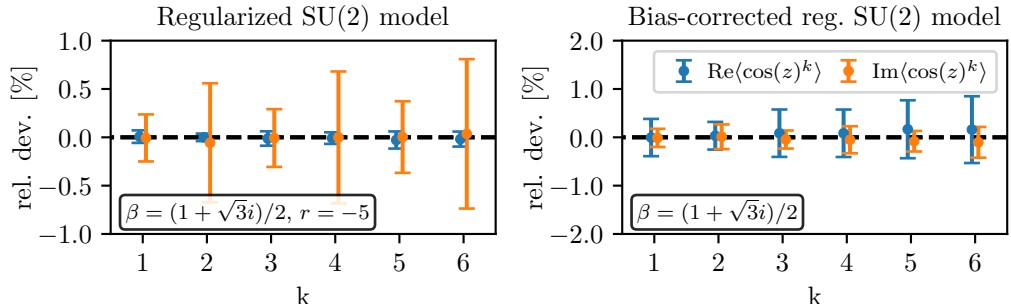

Figure 10: Relative deviation of expectation values of $\langle \cos(z)^k \rangle$ ($k = 1, \ldots, 6$) for the regularized reduced SU(2) Polyakov chain model with weight function $\tilde{\rho} = \rho + R$ with parameters $\beta = (1 + \sqrt{3}i)/2$ and $r = -5$ (*left*) and after the bias correction in Eq. (26) (*right*). The left panel shows that the regularization restores convergence, while the right panel demonstrates the high accuracy of the corrected values in comparison to the original theory. Detailed numerical results are presented in Tables 6 and 7.

In this model, the Jacobian term does not cause a sign problem, so the first drift term is unproblematic. For strong regularization ($|r| \to \infty$), the CL process is pulled toward the real line, while boundary singularities in the regularized action force the thimbles to compactify. This can be seen in the limit when $K(\phi) \approx \partial_\phi \ln[J(\phi)R(\phi)]$. The stationary points are then located at $\phi = \pm \arccos(1/3)$ while the singularities remain at 0 and $\pm\pi$ on the real line. The thimble structure and drift are shown in Fig. 9. Hence, for sufficiently large $|r|$, the regularized model exhibits a single relevant thimble.

In the bottom left panel of Fig. 8 we show the thimble structure of the reduced Polyakov chain model for $\beta = (1 + \sqrt{3}i)/2$ and a regularization force $r = -5$. Up to point symmetry, $\phi \to -\phi$, only one stationary point contributes to the path integral, and the resulting relevant thimble connects $\phi = 0$ and $\phi = \pi$ close to the real line. This compact structure is also respected by the histogram of the thermalized CL trajectory shown in the bottom right panel of Fig. 8. This indicates strongly improved convergence as compared to the unregularized case. This is further supported by the agreement of the exact expectation values of $O_k(\phi) = \cos(\phi)^k$ for the regularized model with our CL results as shown in the left panel of Fig. 10 and listed in Table 6.

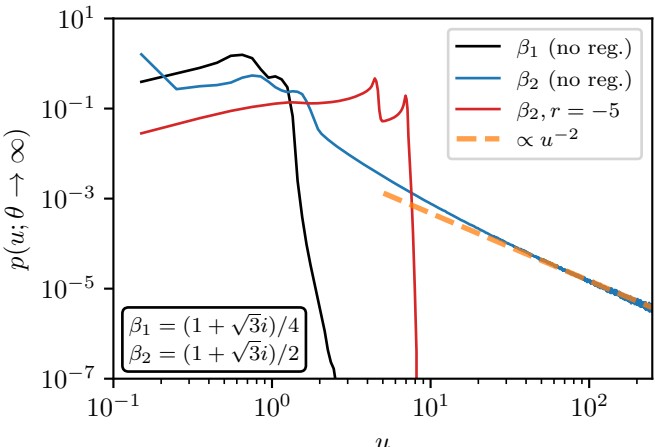

Figure 11: The density $p(u)$ of the drift magnitude $u$ from Eq. (19) for the reduced SU(2) Polyakov chain model for the couplings $\beta_1 = (1+\sqrt{3}i)/4$ and $\beta_2 = (1+\sqrt{3}i)/2$. For $\beta_1$ (gray) the density decays exponentially fast, which confirms the criterion for correct convergence. In contrast, for $\beta_2$ without regularization (blue), the density decreases slowly as a power law (dashed orange line). This is cured after weight regularization (red), which leads to a steep decay of the regularized model and indicates correct convergence.

In complete analogy to the complex cosine model, we extract the expectation values of the original model using a bias correction as discussed in Sec. 4.1. For the determination of the required ratio $Q$ entering the Eq. (26), we take into account the effective action of the original model and, using Eq. (31), obtain for $\mathcal{O} = \sin(\phi)$ the variable

$$\mathcal{O}^*(\phi) = S'_{\text{eff}}(\phi)\sin(\phi) - \cos(\phi). \tag{45}$$

Its expectation value with respect to the original model vanishes due to the Dyson-Schwinger equation. The relative deviation of our thus bias-corrected CL results from the exact values is shown in the right panel of Fig. 10. One finds a remarkably accurate sub-percent deviation that is consistent with the exact results (see also Tab. 7) and confirms the correct convergence of the CL process(es).

The process can be also tested directly regarding correct or wrong convergence. We have analyzed the decay of the density of the drift magnitude, $p(u)$ from Eq. (19), for both coupling values and the effect of regularization, as shown in Fig. 11. For $\beta = (1 + \sqrt{3}i)/4$ (grey curve), the criterion for correctness is satisfied, with $p(u)$ exhibiting the desired exponential decay. In contrast, for $\beta = (1 + \sqrt{3}i)/2$ (blue curve), $p(u)$ decays only according to a power law, failing to satisfy the correctness criterion. However, when the regularization term is introduced with $r = -5$, the density decays at least exponentially, forming a sharp ridge and thereby fulfilling the criterion for correct convergence of the CL process.

In contrast, in Appendix A, we provide an example where for multiple compact relevant thimbles, the criterion is not satisfied. This reinforces that a single compact relevant thimble structure is required.

## 5.2 Weight regularization for the SU(2) link model

We test the weight regularization technique on gauge link models by directly translating the introduced regularization in terms of traces of the Polyakov chain $P$ back to SU(2). We obtain

the regularized action and integral measure

$$S_{\text{reg}}[U] = -\ln\left[e^{-S[U]} + r(\text{Tr}[P]/2 + 1)\right], \quad DU\rho_{\text{reg}}[U] = DU e^{-S_{\text{reg}}[U]}, \quad (46)$$

where $DU$ denotes the Haar-measure for SU(2) and $S[U] = -\beta\,\text{Tr}[P]$. We present the unregularized CL histograms of the thermalized CL process in SU(2) in the top panels of Fig. 12 for $\beta = (1 + \sqrt{3}i)/4$ and $\beta = (1 + \sqrt{3}i)/2$.[7] Similar to the reduced case, the coupling with the smaller magnitude yields a compact and sharp CL histogram following the relevant thimble, while the coupling at the larger magnitude decays slowly and exhibits two contributing thimbles with asymptotic behavior. In the case of $\beta = (1 + \sqrt{3}i)/4$, the expectation values for $\mathcal{O} = (\text{Tr}[U]/2)^k$ from our CL simulations indeed agree accurately with the exact values, as can be seen in Fig. 13 and Table 8. In contrast, CL does not converge correctly for $\beta = (1 + \sqrt{3}i)/2$ (see Table 9). Switching on the regularization term with $r = -5$, the CL process is squeezed toward the real line as shown in the bottom panel of Fig. 12 and converges correctly as presented in the left panel of Fig. 14 and Tab. 10. The bias-corrected expectation for the unregularized model are extracted via the Dyson-Schwinger equation of $\mathcal{O} = \text{Tr}[U]$, yielding

$$\mathcal{O}^*[U] = \beta\left(\text{Tr}[U]^2 - 2\text{Tr}[U^2]\right) - 3\text{Tr}[U], \quad (47)$$

as an observable with vanishing expectation value in the unregularized model to determine the ratio $Q$ in Eq. (26). The extracted results are shown as relative deviations from the exact results in the right panel of Fig. 14. This demonstrates that we can extract the correct expectation values of the original theory to high accuracy using a regularized model where CL works.

We conclude that we can also utilize weight regularizations for the SU(2) link model. This is, in part, expected as we translated the regularized reduced model back to the link formulation. We stress, however, that these two formulations are not necessarily the same in the context of CL. In contrast to the reduced model, where the gauge symmetry has been completely eliminated, the complexified gauge freedom of the link model cannot be fully fixed using the gauge cooling technique. As a result, there are non-compact directions in the complex plane which the CL process eventually explores. This is reminiscent of the redundant degrees of freedom of Lefschetz thimbles in the presence of symmetries. As the correspondence between the high-dimensional thimble structure of the link model and the low-dimensional thimble structure of the reduced model is not fully understood, it is not apparent that an effective regularization for the reduced case improves the convergence properties of the redundant link formulation. Therefore, we emphasize the importance of testing the effect of weight regularization in the presence of gauge symmetries, which may be crucial for the correct convergence of the CL method for lattice gauge theories.

## 6 SU(3) Polyakov chain model

The Polyakov chain model in SU(3) allows us to weigh the Polyakov loop $P$ and its inverse $P^{-1}$ differently in the action, as they yield different numerical values in contrast to SU(2). The action is given by

$$S[P] = -\beta_1\,\text{Tr}[P] - \beta_2\,\text{Tr}[P^{-1}], \quad P \in \text{SU}(3), \quad (48)$$

where $\beta_1 = \beta + \kappa e^{\mu}, \beta_2 = \beta + \kappa e^{-\mu}$. The constants $\beta, \kappa$ can be understood as coupling constants, while $\mu$ may be interpreted as a chemical potential in analogy to the heavy-dense limit

---

[7]We note that the complex Langevin is stabilized using an adaptive time step in addition to the gauge cooling technique with the same simulation parameters as for the unregularized model.

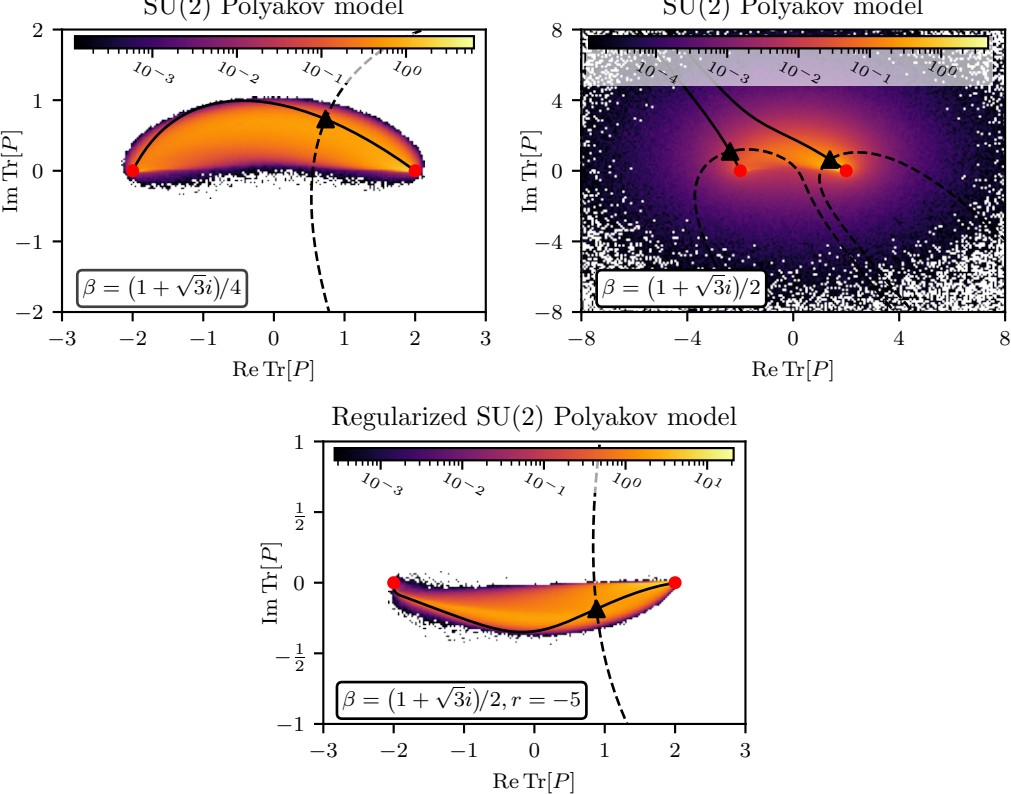

Figure 12: CL histograms with respect to the real and imaginary part of the trace of the Polyakov loop for the SU(2) Polykov chain model Eq. (37) with $\beta = (1 + \sqrt{3}i)/4$ (*top-left*), $\beta = (1 + \sqrt{3}i)/2$ (*top-right*) and with the regularization term $\beta = (1 + \sqrt{3}i)/2, r = -5$ (*bottom*). The green-solid and blue-dashed lines represent thimbles and anti-thimbles of critical points (green triangles) connecting to singular points of the action (green squares). The first coupling exhibits one compact contributing thimble, while the second one leads to asymptotic thimbles that yield dramatic differences in the decay of the histograms. The regularization term leads to faster decay which indicates a better conditioned convergence of the CL process.

of QCD. In SU(3), even for real-valued $\beta, \kappa$, and $\mu$, the model exhibits a sign problem, which results from the complex trace of SU(3) matrices.

We compute expectation values of powers of the trace of the Polyakov loop $\mathcal{O}_k = (\text{Tr}[P]/3)^k$ and its inverse $\tilde{\mathcal{O}}_k = (\text{Tr}[P^{-1}]/3)^k$. Exact values are obtained via direct numerical integration of conjugate classes of the reduced Haar measure, where the Polyakov loop can be represented by $P = \text{diag}(e^{i\phi_1}, e^{i\phi_2}, e^{-i(\phi_1+\phi_2)})$ for $\phi_1, \phi_2 \in [-\pi, \pi]$. The reduced Haar measure for SU(3) is given by [75]

$$dU = d\phi_1 d\phi_2 \sin^2\left(\frac{\phi_1}{2} + \phi_2\right) \sin^2\left(\phi_1 + \frac{\phi_2}{2}\right) \sin^2\left(\frac{\phi_1 - \phi_2}{2}\right). \tag{49}$$

For the reduced model, the thimble analysis is visually less clear, as we have to complexify both angles $\phi_{1,2}$ and, therefore, obtain a four-dimensional space where the thimbles are embedded two-dimensional manifolds. We, thus, proceed by directly applying the knowledge we gained from the previously discussed models to the SU(3) gauge-link model. We design a weight regularization that imposes singularities at the intersection of the boundary of the domain of integration where $\phi_1 = \pm\pi$ or $\phi_2 = \pm\pi$ and the zeros of the Haar measure vanishes. At these points, the trace of the Polyakov loop takes the values $\text{Tr}[P^{\pm 1}] \in \{-1, -1 + 2i, -1 - 2i\}$. Hence,

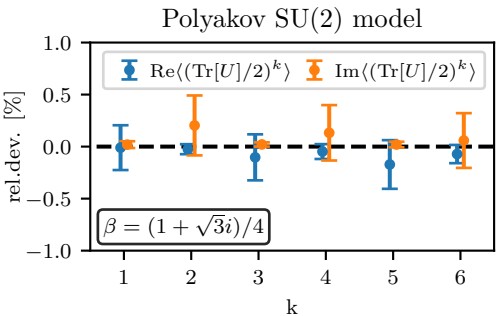

Figure 13: Relative deviation of expectation values of $\langle \cos(z)^k \rangle$ ($k = 1, \ldots, 6$) from the corresponding analytical values for the (unregularized) SU(2) Polyakov chain model with $\beta = (1 + \sqrt{3}i)/4$. For this coupling, the CL process is seen to converge correctly. The values are listed in Table 8.

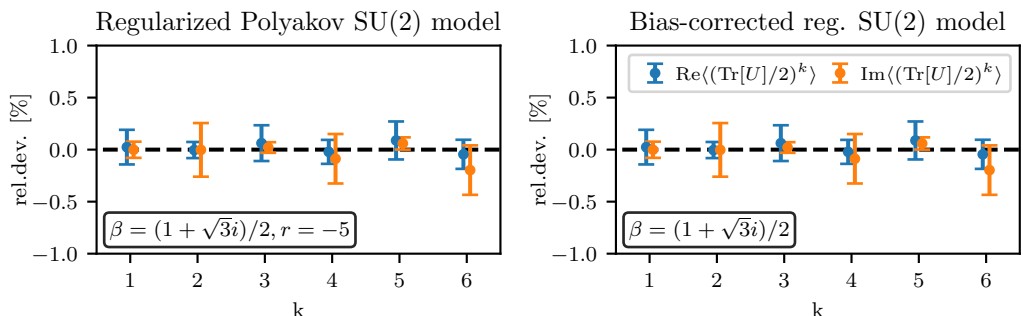

Figure 14: Relative deviation of expectation values of $\langle (\mathrm{Tr}[P]/2)^k \rangle$ ($k = 1, \ldots, 6$) for the regularized (not reduced) SU(2) Polyakov chain model for the weight function $\tilde{\rho} = \rho + R$ with parameters $\beta = (1 + \sqrt{3}i)/2$ and $r = -5$ (*left*) and after the bias correction in Eq. (26) (*right*). The left panel shows that regularization restores convergence, while the right panel demonstrates the high accuracy of the bias-corrected values in comparison to the original theory with weight $\rho$. Detailed numerical results are presented in Tab. 10 and 11.

we employ the regularization

$$R[U] = r\, q(+1)\, q(-1)\,, \tag{50}$$

$$q(\sigma) \equiv (\mathrm{Tr}[P^\sigma] + 1)(\mathrm{Tr}[P^\sigma] + 1 + 2\sigma i)(\mathrm{Tr}[P^\sigma] + 1 - 2\sigma i)\,, \tag{51}$$

where the product in the first line ensures non-negativity on the unitary manifold and $r$ controls the force of the regularization.

In the SU(3) setting, choosing a real gauge coupling $\beta$ makes the sign problem in the Polyakov chain model relatively mild, and gauge cooling helps stabilize the CL process. To make the effects of our weight regularization clearer, we consider an imaginary gauge coupling, $\beta = 0.5i$, with $\mu = 0.5$ and $\kappa = 1$. This setup is similar to the stronger sign problem encountered in real-time simulations.

The matrix-valued drift term for the non-regularized weight function is given by

$$K_j = i\beta_1 t^a \mathrm{Tr}\left[t^a P_j\right] - i\beta_2 t^a \mathrm{Tr}\left[t^a P_j^{-1}\right]\,, \quad P_j = \begin{cases} P\,, & j = 1\,, \\ U_{N_{\mathrm{chain}}} U_1 \ldots U_{j-1}\,, & j = N_{\mathrm{chain}}\,, \\ U_j \ldots U_{N_{\mathrm{chain}}} U_1 \ldots U_{j-1}\,, & \text{otherwise,} \end{cases} \tag{52}$$

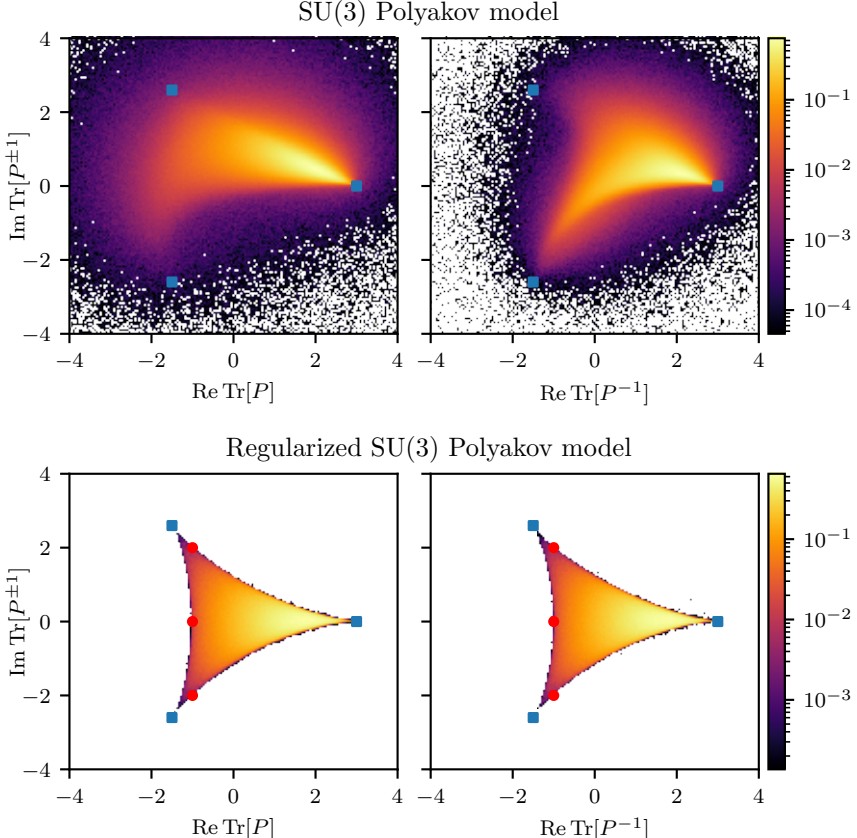

Figure 15: Normalized histograms of the trace of the Polyakov loop (*left panels*) and its inverse (*right panels*) for thermalized CL trajectories in the SU(3) Polyakov chain model with $\beta = 0.5i$, $\mu = 0.5$, and $\kappa = 1.0$. The top panels display results from the unregularized model, exhibiting slow decay in the histogram tails, whereas the bottom panels present sharply defined histograms corresponding to a regularization force set at $r = -25$, indicating improved convergence in the CL method. The red dots represent the singularities at $\text{Tr}\left[P^{\pm 1}\right] \in \{-1, -1+2i, -1-2i\}$ of the reduced Haar measure while the blue squares mark the unit roots $\text{Tr}\left[P^{\pm 1}\right] \in \{3, 3e^{2\pi i/3}, 3e^{4\pi i/3}\}$.

where $t^a = \lambda^a/2$ denote the generators of SU(3) in terms of the Gellman matrices $\lambda^a$ and the index $j$ represents the $j$'th link of the Polyakov chain $P$. For the simulation of the CL process, we use the same simulation parameters as for the SU(2) case, including the adaptive step size and the gauge cooling procedure, which are generalized to SU(3). These methods are also used for the regularized model where the drift changes according to the effective action

$$S_{\text{reg}}[U] = -\ln\{\exp[-S[U]] + R[U]\}, \tag{53}$$

$$K_{\text{reg},j}[U] = \frac{1}{\exp[-S[U]] + R[U]}\left\{\exp[-S[U]]\,K_j[U] + it^a \sum_{\sigma=\pm 1} \sigma w(\sigma)\text{Tr}[t^a P_j^\sigma]\right\}, \tag{54}$$

where we introduced

$$
\begin{aligned}
w(\sigma) = rq(-\sigma)\big[ & (\text{Tr}[P^\sigma]+1+2\sigma i)(\text{Tr}[P^\sigma]+1-2\sigma i) \\
& + (\text{Tr}[P^\sigma]+1)(\text{Tr}[P^\sigma]+1-2\sigma i) \\
& + (\text{Tr}[P^\sigma]+1)(\text{Tr}[P^\sigma]+1+2\sigma i)\big].
\end{aligned} \tag{55}
$$

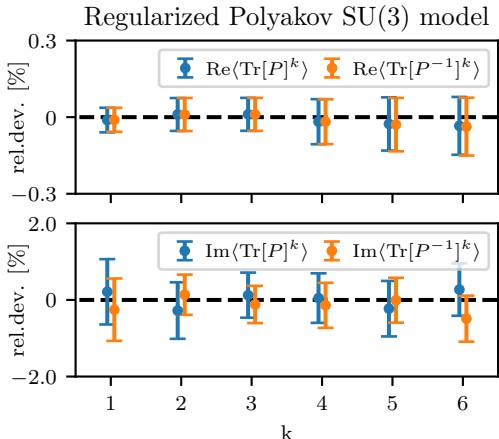

Figure 16: Relative deviation of expectation values for $\text{Tr}[P^{\pm 1}]^k$ in the SU(3) Polyakov chain model with parameters $\beta = 0.5i$, $\mu = 0.5$, and $\kappa = 1.0$, regularization applied at $r = -25$. The regularization leads to correct results, while without this regularization, we obtain wrong results. The numerical values for the original and regularized model are presented in the Tables 12 to 15.

In order to correct for the bias that stems from our weight regularization, we follow the method in Eq. (26). As in the other models discussed previously, we determine the required ratio $Q$ using Eq. (29). The Dyson-Schwinger equations of the traced Polyakov loop and its inverse with respect to the original action $S$ are given by

$$\langle \mathcal{O}^* \rangle \equiv \left\langle 2\beta_1 \left( \text{Tr}[P^2] - \frac{\text{Tr}[P]^2}{3} \right) - 2\beta_2 \left( N - \frac{\text{Tr}[P]\text{Tr}[P^{-1}]}{3} \right) + \frac{16}{3}\text{Tr}[P] \right\rangle_\rho = 0, \qquad (56)$$

$$\langle \tilde{\mathcal{O}}^* \rangle \equiv \left\langle 2\beta_1 \left( N - \frac{\text{Tr}[P]\text{Tr}[P^{-1}]}{3} \right) - 2\beta_2 \left( \text{Tr}[P^{-2}] - \frac{\text{Tr}[P^{-1}]^2}{3} \right) - \frac{16}{3}\text{Tr}[P^{-1}] \right\rangle_\rho = 0. \quad (57)$$

We can then compare our bias-corrected CL results with the exact values of observables.

In the top panels of Fig. 15, we show the normalized histograms of the trace of the Polyakov loop and its inverse for thermalized CL processes, where no regularization was applied, though gauge cooling was used to reduce the gauge freedom and minimize the unitarity norm. We observe that these histograms decay only slowly. This results in divergent expectation values for $\mathcal{O}_k = \text{Tr}[P]^k$ with $k = 1, \ldots, 6$ (see Tables 12 and 13). The bottom panels of Fig. 15 show the resulting histograms when the regularization term is activated with $r = -25$. This produces sharply defined histograms that show a $\mathbb{Z}_3$-structure. This regularization leads to correct convergence, as shown in Fig. 16. The numerical values are presented in Tables 14 and 15. Following the same approach as for the models before, we correct the distortion in the original expectation values caused by our regularization by using our procedure based on the Dyson-Schwinger equations for $\text{Tr}[P^\pm]$ in Eq. (56). As shown in Fig. 17 for the relative deviation from the exact values (see Tables 16 to 19), the bias-corrected results agree with the original model. This demonstrates that the CL method can accurately extract expectation values in the SU(3) model.

We emphasize that statistical uncertainties are highly sensitive to the precision with which the ratio $Q$ is determined. In principle, the Dyson-Schwinger equation for any suitable observable may be employed, but a variance-reduced combination where multiple observables are used to counteract systematic and statistical uncertainties could reduce the final error. The shown consistency across several powers $\mathcal{O}_k$ provides strong evidence for the viability of this approach.

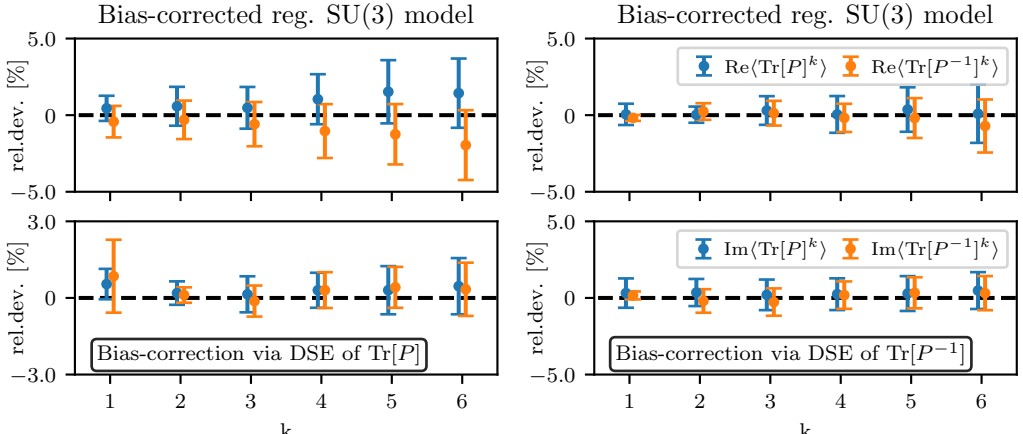

Figure 17: Relative deviation of corrected expectation values for the real (*top*) and the imaginary (*bottom*) parts of $\text{Tr}[P^{\pm 1}]^k$ in the regularized SU(3) Polyakov chain model with parameters $\beta = 0.5i$, $\mu = 0.5$, $\kappa = 1.0$, and regularization force $r = -25$. The corrections were computed using the DSE of $\text{Tr}[P]$ (*left*) and $\text{Tr}[P^{-1}]$ (*right*). Both DSEs for the respective observables lead to statistical consistency and negligible deviation when compared to the exact result for the original model.

# 7 Discussion and conclusion

In this work, we investigated the use of weight regularization to modify the Lefschetz thimble structure of specific theories, with the goal of ensuring the correct convergence of CL simulations. Our analysis of the established criterion for correctness of CL and its connection to the thimble structure provides strong empirical evidence that CL converges correctly when a unique compact Lefschetz thimble contributes to the path integral.

In particular, we have applied the weight regularization method to several toy models, including the complex cosine model and different formulations of the Polyakov chain model with complex couplings for the gauge groups SU(2) and SU(3). These models exhibit a severe sign problem and violate the criterion for correct convergence of the CL method by Nagata et al. [57], as we explicitly demonstrate for some of the models. By introducing weight regularization, we successfully restore the correctness criteria and reliably extract accurate expectation values for these theories, resolving previous CL failures. Importantly, the bias introduced by the regularization term is systematically removed in a simple procedure using the fact that the Dyson-Schwinger equations hold in the original theory but not in the regularized system, which enables a precise bias correction. Our approach involves designing an additive term that modifies the original weight function. The choice of this regularization is not unique, but we motivate its specific form based on the resulting Lefschetz thimble structure. In particular, we aim to achieve a single compact relevant thimble by introducing positive-definite regularization terms that vanish at the boundaries of the integration domain. We manage to find suitable regularization terms for each of the considered models and emphasize that the detailed choice of the weight regularization term depends heavily on the underlying theory and the coordinate system in which the model is formulated. Specifically for the non-Abelian gauge models, we find that utilizing the zeros of the Haar measure helps to construct suitable regularizations.

Consistent with previous studies, our numerical evidence indicates a strong connection between the success and failure of CL and the associated thimble structure of the system. Our study strongly suggests that a single compact thimble is necessary to obtain correct conver-

gence for theories with a compact real integration domain. Even for a configuration of multiple compact relevant thimbles, we have found that we exhibit wrong convergence (see Appendix A). In the course of this investigation, we have not encountered a single counterexample to this conjecture. A precise analytic formulation of CL's correctness criteria in terms of Lefschetz thimbles is highly desirable.

A generalization of these ideas to higher-dimensional field theories would be highly desirable. However, a direct extension appears challenging since the construction of suitable additive regularization terms is unclear. There are two major difficulties: First, in general, there is no systematic way to reliably study Lefschetz thimbles in higher-dimensional settings. Hence, it is not straightforward to motivate a regularization term that deforms the thimbles adequately. Despite this obstacle, one may promote successful regularization terms from lower-dimensional theories. The second, potentially more problematic impediment is that additive regularizations appear to be incompatible with lattice models. Either the regularized lattice field theory becomes non-local or the bias correction becomes intractable (see Appendix B).

Despite these complications, we may utilize the idea of deforming the relevant thimbles of the theory by employing *multiplicative* instead of additive regularization terms where the weight function $\rho$ is adapted according to $\rho \rightarrow \rho R$. The effect of these terms can be bias-corrected by reweighting the result of the regularized weight function. Unfortunately, this approach suffers from the well-known overlap problem associated with reweighting methods.

Another promising approach involves using CL kernels to effectively transform the complex Langevin equation and, thus, enhance its convergence towards the desired probability distribution. Recently, several interesting applications of this method have emerged [40–42,52,53,76]. However, the relationship between thimbles and kernels remains an open question. Understanding this connection is the focus of ongoing research and could enable the design of field-dependent kernels that adjust the thimble structure. Such insights may provide a constructive framework for systematically designing these kernels for field theories where CL is known to fail.

While the weight regularization technique may not be directly applicable to high-dimensional field theories, we have provided evidence supporting a connection between the thimble structure of a given theory and the convergence criteria of CL, as previously conjectured. By designing tailored regularization terms to achieve a compact single-thimble structure, we successfully solved both the complex cosine model and the Polyakov chain model for different gauge groups. This work, therefore, paves the way for developing stabilization techniques that leverage the thimble structure as a guiding principle. Our ongoing efforts are focused on designing kernels suited for SU(N) field theories, with the aim of extending these methods to real-time Yang-Mills theory and finite-density QCD in the future.

## Acknowledgments

The computational results presented here have been achieved using the Vienna Scientific Cluster (VSC).

**Funding information**   This work is funded in whole or in part by the Austrian Science Fund (FWF) under the projects [10.55776/P34455] and W 1252. D.M. acknowledges additional support from project P 34764.

# A Multiple relevant compact thimbles

We emphasize in this work, it is important to have a single relevant compact thimble for the CL method to converge correctly. Here, we show an example where CL fails due to the existence of multiple relevant compact thimbles of the system.

We revisit the reduced SU(2) Polyakov loop model and regularize it with a regularization term according to

$$\tilde{\rho}(\phi) = \sin^2(\phi)\left[e^{2\beta\cos(\phi)} + r\sin^2(\phi)\right]. \tag{A.1}$$

For this regularization, we observe a curious phenomenon for $\beta = (1 + \sqrt{3}i)/2$. The thimble structure is compressed towards the real line for $|r| \to \infty$ as intended but the regularized model exhibits multiple relevant stationary points. In the left panel of Fig. 18, we show the thimble structure of the regularized model for $r = 0.5$ where three thimbles are relevant in the left and right half-planes and connect to singularities of the effective action such that the resulting structure stays compact. Despite the compactness, we show in the right panel of Fig. 18 that the criterion of correctness for CL is not satisfied as the density of the drift magnitude decays only like a power law and, thus, not fast enough.

We observe that the characteristic structure involving multiple relevant, compact thimbles persists even for larger values of $|r|$. Alongside this, the drift magnitude density continues to exhibit a slow, power-law-like decay. This slow decay can be attributed to singularities located in the bulk of the complex plane, as the probability density does not decay rapidly enough in their neighborhood. Consequently, the regularization term $R(\phi) = r\sin^2(\phi)$ proves inadequate for ensuring correct convergence, as reliable results can only be expected asymptotically in the extreme limit of $|r| \to \infty$. In practical terms, this renders the regularization unsuitable for addressing the model's convergence issues within a feasible numerical framework.

We emphasize, however, that this regularized model provides valuable insights with regard to the conjecture for the connection between Lefschetz thimbles and CL's criterion of correctness: Although all relevant thimbles are compact, we still observe wrong convergence. This is also confirmed by directly simulating the CL equation for this model. Hence, this underpins that not more than a single thimble may be indeed necessary in order to guarantee correct convergence.

# B Additive regularization and lattice field theories

The weight functions of lattice field theories are generally of the form

$$\rho = \prod_x \rho_x = e^{-S} = e^{-\sum_x \mathcal{L}_x}, \tag{B.1}$$

where $x$ is the index of a specific space-time point and $\mathcal{L}_x$ can be interpreted as the discrete analog of the Lagrange density. The degrees of freedom, e.g. some lattice field values $\phi_x$, couple only locally via $\mathcal{L}_x$. For example, a $\phi^4$-theory in $D$ dimensions would have

$$\mathcal{L}_x = -\frac{1}{2}\phi_x\sum_{\mu=1}^{D}(\phi_{x+\hat{\mu}} + \phi_{x-\hat{\mu}} - 2\phi_x) + \frac{1}{2}m^2\phi_x^2 + \frac{1}{4!}\lambda^4\phi_x^4, \tag{B.2}$$

with a mass $m$ and a coupling constant $\lambda$.

To regularize such a lattice field theory, we could choose the regularization term to act globally, e.g.,

$$\tilde{\rho} = \prod_x \rho_x + \prod_x R_x = \rho + R, \tag{B.3}$$

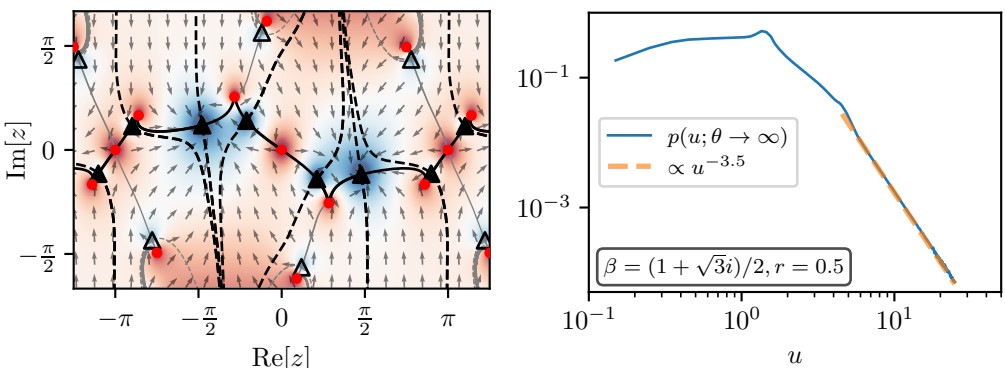

Figure 18: Thimble structures *(left)* and the density of the drift magnitude $p(u; \theta \to \infty)$ *(right)* of the regularized, reduced SU(2) Polyakov chain model for the coupling $\beta = (1 + i\sqrt{3})/2$ and regularization force $r = 0.5$. The considered model is regularized using $R(\phi) = r\sin^2(\phi)$. We use the same conventions as in Fig. 1 for the thimble structure and observe that multiple compact thimbles contribute in this setting. Although all relevant thimbles are compact, the criterion of correctness is not satisfied as the drift magnitude decays only like a power law.

where $R_x$ is a local regularization term. The issue with a global additive regularization is that the effective action of the regularized system $\tilde{S} = -\ln\tilde{\rho}$ and the associated drift term become highly non-local. Apart from the practical issues of solving a non-local Langevin equation numerically, the interpretation of the regularized non-local theory as a lattice field theory in the usual sense is unclear.

To avoid non-locality, it is possible to regularize the weight function for different space-time points individually, e.g.,

$$\rho = \prod_x \rho_x \to \prod_x (\rho_x + R_x) = \tilde{\rho}. \tag{B.4}$$

While the effective action and drift are local in this case, the complexity of the bias correction increases with the lattice size. To see this, consider a non-interacting lattice model with merely two lattice sites and field values $\phi_1$, $\phi_2$ regularized by

$$\tilde{\rho} = (e^{-\mathcal{L}_1} + R_1(\phi_1))(e^{-\mathcal{L}_2} + R_2(\phi_2)). \tag{B.5}$$

Following the bias-correction procedure, we decompose the partition function

$$Z_{\tilde{\rho}} = Z_\rho + Z_{(1)} + Z_{(2)} + Z_R, \tag{B.6}$$

$$Z_{(1)} = \int d\phi_1 d\phi_2 \, e^{-S_1} R_2, \quad Z_{(2)} = \int d\phi_1 d\phi_2 \, e^{-S_2} R_1, \quad Z_R = \int d\phi_1 d\phi_2 \, R_1 R_2. \tag{B.7}$$

Similarly, the expectation value of an observable admits the decomposition

$$\langle \mathcal{O} \rangle_{\tilde{\rho}} = \frac{Z_\rho}{Z_{\tilde{\rho}}} \langle \mathcal{O} \rangle_\rho + \frac{Z_{(1)}}{Z_{\tilde{\rho}}} \langle \mathcal{O} \rangle_{(1)} + \frac{Z_{(2)}}{Z_{\tilde{\rho}}} \langle \mathcal{O} \rangle_{(2)} + \frac{Z_R}{Z_{\tilde{\rho}}} \langle \mathcal{O} \rangle_R. \tag{B.8}$$

Compared to the single-variable case, see Eq. (26), we now require the computation of $\langle \mathcal{O} \rangle_{(1)}$ and $\langle \mathcal{O} \rangle_{(2)}$, in addition to $\langle \mathcal{O} \rangle_{\tilde{\rho}}$ and $\langle \mathcal{O} \rangle_R$. To correct for the bias, the ratios $Z_{(1)}/Z_{\tilde{\rho}}$ and $Z_{(2)}/Z_{\tilde{\rho}}$ have to be determined as well.

This argument can be generalized to arbitrary lattices $\Omega$. In analogy to Eq. (B.8), the bias correction includes a separate term $\langle O \rangle_{\tilde{\rho}_V} \frac{Z_{\tilde{\rho}_V}}{Z_{\tilde{\rho}_\Omega}}$ for every subset $V \subseteq \Omega$, with the product

$$\tilde{\rho}_V \equiv \prod_{x \in V} \rho_x \prod_{x \in \Omega \setminus V} R_x . \tag{B.9}$$

This complexity is aggravated when $\Omega$ scales up as this increases the number of coefficients that need to be determined and renders a direct bias correction intractable.

# C  Detailed results for the cosine model

Table 1: Numerical results for the complex cosine model at coupling $\beta = 0.5$ for the observable $\mathcal{O}_k = \cos(z)^k$, computed using the CL method. The table compares the real and imaginary parts of the exact results (columns 2 and 4) with the corresponding results from the CL simulation (columns 3 and 5). For this parameter set, the CL method fails to converge correctly. It is noteworthy that the observables for $k > 2$ are known to diverge.

| Order $k$ | $\mathrm{Re}\langle\mathcal{O}_k\rangle_\rho^{\mathrm{exact}}$ | $\mathrm{Re}\langle\mathcal{O}_k\rangle_\rho$ | $\mathrm{Im}\langle\mathcal{O}_k\rangle_\rho^{\mathrm{exact}}$ | $\mathrm{Im}\langle\mathcal{O}_k\rangle_\rho$ |
|---|---|---|---|---|
| 1 | 0 | -0.38(2) | -0.258153 | -12.22(6) |
| 2 | 0.483695 | $-1.541(7)\times10^4$ | 0 | $1.15(9)\times10^3$ |
| 3 | 0 | $2.2(1)\times10^6$ | -0.192932 | $1.45(2)\times10^7$ |
| 4 | 0.358716 | $5.3(2)\times10^9$ | 0 | $-3.1(3)\times10^9$ |
| 5 | 0 | $-2.6(3)\times10^{12}$ | -0.160492 | $1.39(5)\times10^{13}$ |
| 6 | 0.297246 | $3.51(5)\times10^{16}$ | 0 | $-1.3(8)\times10^{15}$ |

Table 2: Numerical results for the regularized complex cosine model at coupling $\beta = 0.5$ and regularization force $r = 0.5$ for the observable $\mathcal{O}_k = \cos(z)^k$, computed using the CL method. The table compares the real and imaginary parts of the exact results for the regularized model $\rho + R$ (columns 2 and 4) with the corresponding results from the CL simulation (columns 3 and 5). In contrast to the unregularized model, CL converges correctly and agrees with the exact results.

| Order $k$ | $\mathrm{Re}\langle\mathcal{O}_k\rangle_{\rho+R}^{\mathrm{exact}}$ | $\mathrm{Re}\langle\mathcal{O}_k\rangle_{\rho+R}$ | $\mathrm{Im}\langle\mathcal{O}_k\rangle_{\rho+R}^{\mathrm{exact}}$ | $\mathrm{Im}\langle\mathcal{O}_k\rangle_{\rho+R}$ |
|---|---|---|---|---|
| 1 | 0.313915 | 0.3138(3) | 0.028421 | 0.0284(2) |
| 2 | 0.466760 | 0.4667(1) | 0.004935 | 0.00494(7) |
| 3 | 0.243825 | 0.2437(2) | 0.019872 | 0.0199(1) |
| 4 | 0.339386 | 0.3392(1) | 0.005288 | 0.00529(8) |
| 5 | 0.207440 | 0.2073(2) | 0.015846 | 0.0158(1) |
| 6 | 0.277514 | 0.2774(1) | 0.005195 | 0.00520(7) |

Table 3: Bias-corrected results for the regularized complex cosine model at coupling $\beta = 0.5$ and regularization force $r = 0.5$ for the observable $\mathcal{O}_k = \cos(z)^k$, computed using the CL method. The table compares the real and imaginary parts of the exact results for the original model $\rho$ (columns 2 and 4) with the corresponding bias-corrected results for $\rho + R$ from the CL simulation (columns 3 and 5). The bias-corrected results agree with the exact results.

| Order $k$ | $\mathrm{Re}\langle\mathcal{O}_k\rangle_\rho^{\mathrm{exact}}$ | $\mathrm{Re}\langle\mathcal{O}_k\rangle_{\rho+R}^{\mathrm{bias-corr.}}$ | $\mathrm{Im}\langle\mathcal{O}_k\rangle_\rho^{\mathrm{exact}}$ | $\mathrm{Im}\langle\mathcal{O}_k\rangle_{\rho+R}^{\mathrm{bias-corr.}}$ |
|---|---|---|---|---|
| 1 | 0 | 0.000(3) | -0.258153 | -0.2581(5) |
| 2 | 0.483695 | 0.484(1) | 0 | -0.0001(2) |
| 3 | 0 | 0.000(2) | -0.192932 | -0.1930(4) |
| 4 | 0.358716 | 0.359(1) | 0 | -0.0001(2) |
| 5 | 0 | 0.000(2) | -0.160492 | -0.1606(4) |
| 6 | 0.297246 | 0.298(1) | 0 | -0.0000(2) |

# D   Detailed results for the reduced SU(2) Polyakov chain model

Table 4: Numerical results for the reduced SU(2) Polyakov chain model at coupling $\beta = (1 + \sqrt{3})/4$ for the observable $\mathcal{O}_k = \cos(z)^k$. At this coupling, CL converges correctly and yields reliable results. The same notation as in Table 1 is used.

| Order $k$ | $\mathrm{Re}\langle\mathcal{O}_k\rangle_\rho^{\mathrm{exact}}$ | $\mathrm{Re}\langle\mathcal{O}_k\rangle_\rho$ | $\mathrm{Im}\langle\mathcal{O}_k\rangle_\rho^{\mathrm{exact}}$ | $\mathrm{Im}\langle\mathcal{O}_k\rangle_\rho$ |
|---|---|---|---|---|
| 1 | 0.135717 | 0.1356(2) | 0.215905 | 0.2159(2) |
| 2 | 0.235486 | 0.2355(1) | 0.028747 | 0.0287(1) |
| 3 | 0.065165 | 0.0652(1) | 0.108133 | 0.1081(1) |
| 4 | 0.114019 | 0.11407(9) | 0.021420 | 0.0215(1) |
| 5 | 0.039721 | 0.03975(9) | 0.067643 | 0.0676(1) |
| 6 | 0.069842 | 0.06989(8) | 0.015995 | 0.01603(9) |

Table 5: Numerical results for the reduced SU(2) Polyakov chain model at coupling $\beta = (1 + \sqrt{3})/2$ for the observable $\mathcal{O}_k = \cos(z)^k$. CL converges wrongly and appears to diverge for sufficiently large $k$. To obtain finite results a cutoff at $|z| < 6\pi$ is imposed. The same notation as in Table 1 is used.

| Order $k$ | $\mathrm{Re}\langle\mathcal{O}_k\rangle_\rho^{\mathrm{exact}}$ | $\mathrm{Re}\langle\mathcal{O}_k\rangle_\rho$ | $\mathrm{Im}\langle\mathcal{O}_k\rangle_\rho^{\mathrm{exact}}$ | $\mathrm{Im}\langle\mathcal{O}_k\rangle_\rho$ |
|---|---|---|---|---|
| 1 | 0.339351 | -2.60(1) | 0.410601 | 5.53(2) |
| 2 | 0.212100 | -968(6) | 0.132879 | -1615(8) |
| 3 | 0.147098 | $4.48(2)\times10^5$ | 0.212077 | $-1.6(3)\times10^4$ |
| 4 | 0.094335 | $-1.5(1)\times10^7$ | 0.097673 | $4.0(1)\times10^7$ |
| 5 | 0.083524 | $1.21(4)\times10^{10}$ | 0.134791 | $2.58(6)\times10^{10}$ |
| 6 | 0.054030 | $-1.99(2)\times10^{13}$ | 0.072252 | $-6(3)\times10^{11}$ |

Table 6: Numerical results for the regularized reduced SU(2) Polyakov chain model at the coupling $\beta = (1 + \sqrt{3})/2$ and regularization force $r = -5$ for the observable $\mathcal{O}_k = \cos(z)^k$. Unlike for the original model, CL converges correctly when the regularization is imposed and agrees with the exact results. The same notation as in Table 2 is used.

| Order $k$ | Re$\langle\mathcal{O}_k\rangle_{\rho+R}^{\text{exact}}$ | Re$\langle\mathcal{O}_k\rangle_{\rho+R}$ | Im$\langle\mathcal{O}_k\rangle_{\rho+R}^{\text{exact}}$ | Im$\langle\mathcal{O}_k\rangle_{\rho+R}$ |
|---|---|---|---|---|
| 1 | 0.275920 | 0.2759(2) | -0.073871 | -0.0739(2) |
| 2 | 0.268989 | 0.2690(1) | -0.017381 | -0.0174(1) |
| 3 | 0.142204 | 0.1422(1) | -0.035805 | -0.0358(1) |
| 4 | 0.139403 | 0.13941(8) | -0.012502 | -0.01250(9) |
| 5 | 0.090431 | 0.09046(8) | -0.021913 | -0.02191(8) |
| 6 | 0.089003 | 0.08902(7) | -0.009110 | -0.00911(7) |

Table 7: Bias-corrected results for the regularized reduced SU(2) Polyakov chain model at the coupling $\beta = (1 + \sqrt{3})/2$ and regularization force $r = -5$ for the observable $\mathcal{O}_k = \cos(z)^k$. The bias correction procedure yields results that are in agreement with the exact results. The same notation as in Table 3 is used.

| Order $k$ | Re$\langle\mathcal{O}_k\rangle_{\rho}^{\text{exact}}$ | Re$\langle\mathcal{O}_k\rangle_{\rho+R}^{\text{bias-corr.}}$ | Im$\langle\mathcal{O}_k\rangle_{\rho}^{\text{exact}}$ | Im$\langle\mathcal{O}_k\rangle_{\rho+R}^{\text{bias-corr.}}$ |
|---|---|---|---|---|
| 1 | 0.339351 | 0.339(1) | 0.410601 | 0.4106(8) |
| 2 | 0.212100 | 0.2120(6) | 0.132879 | 0.1329(3) |
| 3 | 0.147098 | 0.1470(7) | 0.212077 | 0.2122(4) |
| 4 | 0.094335 | 0.0943(5) | 0.097673 | 0.0977(3) |
| 5 | 0.083524 | 0.0834(5) | 0.134791 | 0.1349(3) |
| 6 | 0.054030 | 0.0539(4) | 0.072252 | 0.0723(2) |

# E  Detailed results for the SU(2) Polyakov chain model

Table 8: Numerical results for the SU(2) Polyakov chain link model at the coupling $\beta = (1 + \sqrt{3})/4$ for the observables $\mathcal{O}_k = (\text{Tr}[P]/2)^k$, $k = 1, \ldots, 6$. The same notation as in Table 1 is used.

| Order $k$ | Re$\langle\mathcal{O}_k\rangle_{\rho}^{\text{exact}}$ | Re$\langle\mathcal{O}_k\rangle_{\rho}$ | Im$\langle\mathcal{O}_k\rangle_{\rho}^{\text{exact}}$ | Im$\langle\mathcal{O}_k\rangle_{\rho}$ |
|---|---|---|---|---|
| 1 | 0.135717 | 0.1357(3) | 0.215905 | 0.21586(7) |
| 2 | 0.235486 | 0.2355(1) | 0.028747 | 0.02869(8) |
| 3 | 0.065165 | 0.0652(1) | 0.108133 | 0.10811(2) |
| 4 | 0.114019 | 0.11407(8) | 0.021420 | 0.02139(6) |
| 5 | 0.039721 | 0.03979(9) | 0.067643 | 0.06763(2) |
| 6 | 0.069842 | 0.06989(6) | 0.015995 | 0.01599(4) |

Table 9: Numerical results for the SU(2) Polyakov chain link model at coupling $\beta = (1 + \sqrt{3})/2$ for the observables $\mathcal{O}_k = (\mathrm{Tr}[P]/2)^k$, $k = 1, \ldots, 6$. CL converges wrongly for this coupling, and to produce finite results, a cutoff at $|\mathrm{Tr}[P]| = 8$ was imposed. The same notation as in Table 1 is used.

| Order $k$ | Re$\langle\mathcal{O}_k\rangle_\rho^{\mathrm{exact}}$ | Re$\langle\mathcal{O}_k\rangle_\rho$ | Im$\langle\mathcal{O}_k\rangle_\rho^{\mathrm{exact}}$ | Im$\langle\mathcal{O}_k\rangle_\rho$ |
|---|---|---|---|---|
| 1 | 0.339351 | 0.3479(3) | 0.410601 | 0.3515(1) |
| 2 | 0.212100 | 0.2466(2) | 0.132879 | 0.1327(3) |
| 3 | 0.147098 | 0.1522(4) | 0.212077 | 0.1938(5) |
| 4 | 0.094335 | 0.119(1) | 0.097673 | 0.097(1) |
| 5 | 0.083524 | 0.089(3) | 0.134791 | 0.126(3) |
| 6 | 0.054030 | 0.08(1) | 0.072252 | 0.07(1) |

Table 10: Numerical results for the regularized SU(2) Polyakov chain link model at coupling $\beta = (1 + \sqrt{3})/2$ with a regularization force of $r = -5$ for the observables $\mathcal{O}_k = (\mathrm{Tr}[P]/2)^k$, $k = 1, \ldots, 6$. In contrast to the original model, CL converges correctly when the regularization is switched on. The same notation as in Table 2 is used.

| Order $k$ | Re$\langle\mathcal{O}_k\rangle_{\rho+R}^{\mathrm{exact}}$ | Re$\langle\mathcal{O}_k\rangle_{\rho+R}$ | Im$\langle\mathcal{O}_k\rangle_{\rho+R}^{\mathrm{exact}}$ | Im$\langle\mathcal{O}_k\rangle_{\rho+R}$ |
|---|---|---|---|---|
| 1 | 0.275920 | 0.2759(5) | -0.073871 | -0.07387(6) |
| 2 | 0.268989 | 0.2690(2) | -0.017381 | -0.01738(4) |
| 3 | 0.142204 | 0.1421(2) | -0.035805 | -0.03581(2) |
| 4 | 0.139403 | 0.1394(2) | -0.012502 | -0.01249(3) |
| 5 | 0.090431 | 0.0904(2) | -0.021913 | -0.02193(1) |
| 6 | 0.089003 | 0.0890(1) | -0.009110 | -0.00909(2) |

Table 11: Bias-corrected results for the regularized SU(2) Polyakov chain link model at coupling $\beta = (1 + \sqrt{3})/2$ with a regularization force of $r = -5$ for the observables $\mathcal{O}_k = (\mathrm{Tr}[P]/2)^k$, $k = 1, \ldots, 6$. The bias correction procedure yields results that are in agreement with the exact results. The same notation as in Table 3 is used.

| Order $k$ | Re$\langle\mathcal{O}_k\rangle_\rho^{\mathrm{exact}}$ | Re$\langle\mathcal{O}_k\rangle_{\rho+R}^{\mathrm{bias-corr.}}$ | Im$\langle\mathcal{O}_k\rangle_\rho^{\mathrm{exact}}$ | Im$\langle\mathcal{O}_k\rangle_{\rho+R}^{\mathrm{bias-corr.}}$ |
|---|---|---|---|---|
| 1 | 0.275920 | 0.2759(5) | -0.073871 | -0.07387(6) |
| 2 | 0.268989 | 0.2690(2) | -0.017381 | -0.01738(4) |
| 3 | 0.142204 | 0.1421(2) | -0.035805 | -0.03581(2) |
| 4 | 0.139403 | 0.1394(2) | -0.012502 | -0.01249(3) |
| 5 | 0.090431 | 0.0904(2) | -0.021913 | -0.02193(1) |
| 6 | 0.089003 | 0.0890(1) | -0.009110 | -0.00909(2) |

## F  Detailed results for the SU(3) Polyakov chain model

Table 12: Numerical results for the SU(3) Polyakov chain link model at coupling $\beta = 0.5i$, $\kappa = 1$ and $\mu = 0.5$ for the observables $\mathcal{O}_k = (\mathrm{Tr}[P]/3)^k$, $k = 1,\ldots,6$. CL converges wrongly for this coupling, and to produce finite results, a cutoff at $|\mathrm{Tr}[P]| = 9$ was imposed. The same notation as in Table 1 is used.

| Order $k$ | $\mathrm{Re}\langle\mathcal{O}_k\rangle_\rho^{\mathrm{exact}}$ | $\mathrm{Re}\langle\mathcal{O}_k\rangle_\rho$ | $\mathrm{Im}\langle\mathcal{O}_k\rangle_\rho^{\mathrm{exact}}$ | $\mathrm{Im}\langle\mathcal{O}_k\rangle_\rho$ |
|---|---|---|---|---|
| 1 | 0.347315 | 0.3512(3) | 0.259843 | 0.2486(2) |
| 2 | 0.152430 | 0.1566(6) | 0.139517 | 0.1432(7) |
| 3 | 0.082942 | 0.09(1) | 0.097470 | 0.11(2) |
| 4 | 0.047047 | -0.1(3) | 0.070313 | 1.1(5) |
| 5 | 0.027324 | $1(1)\times10^1$ | 0.047738 | $1(1)\times10^1$ |
| 6 | 0.017663 | $2(3)\times10^2$ | 0.035714 | $4(3)\times10^2$ |

Table 13: Numerical results for the SU(3) Polyakov chain link model at the coupling $\beta = 0.5i$, $\kappa = 1$ and $\mu = 0.5$ for the observables $\mathcal{O}_k = \left(\mathrm{Tr}\left[P^{-1}\right]/3\right)^k$, $k = 1,\ldots,6$. CL converges wrongly for this coupling, and to produce finite results, a cutoff at $|\mathrm{Tr}\left[P^{-1}\right]| = 9$ was imposed. The same notation as in Table 1 is used.

| Order $k$ | $\mathrm{Re}\langle\mathcal{O}_k\rangle_\rho^{\mathrm{exact}}$ | $\mathrm{Re}\langle\mathcal{O}_k\rangle_\rho$ | $\mathrm{Im}\langle\mathcal{O}_k\rangle_\rho^{\mathrm{exact}}$ | $\mathrm{Im}\langle\mathcal{O}_k\rangle_\rho$ |
|---|---|---|---|---|
| 1 | 0.419999 | 0.4180(3) | 0.156268 | 0.1548(2) |
| 2 | 0.191396 | 0.1929(4) | 0.156389 | 0.1533(7) |
| 3 | 0.096932 | 0.10(1) | 0.103684 | 0.11(1) |
| 4 | 0.053733 | -0.3(3) | 0.069514 | 0.2(1) |
| 5 | 0.032540 | 3(4) | 0.051796 | 3(5) |
| 6 | 0.020094 | $5(9)\times10^1$ | 0.037707 | $-1(1)\times10^2$ |

Table 14: Numerical results for the regularized SU(3) Polyakov chain link model at the coupling $\beta = 0.5i$, $\kappa = 1$ and $\mu = 0.5$ with a regularization force $r = -25$ for the observables $\mathcal{O}_k = (\mathrm{Tr}[P]/3)^k$, $k = 1,\ldots,6$. In contrast to the original model, the regularization term leads to correct convergence. The same notation as in Table 2 is used.

| Order $k$ | $\mathrm{Re}\langle\mathcal{O}_k\rangle_{\rho+R}^{\mathrm{exact}}$ | $\mathrm{Re}\langle\mathcal{O}_k\rangle_{\rho+R}$ | $\mathrm{Im}\langle\mathcal{O}_k\rangle_{\rho+R}^{\mathrm{exact}}$ | $\mathrm{Im}\langle\mathcal{O}_k\rangle_{\rho+R}$ |
|---|---|---|---|---|
| 1 | 0.253719 | 0.2537(1) | 0.006856 | 0.00684(6) |
| 2 | 0.115097 | 0.11509(7) | 0.003287 | 0.00330(2) |
| 3 | 0.071673 | 0.07166(5) | 0.001845 | 0.00184(1) |
| 4 | 0.039406 | 0.03941(3) | 0.001316 | 0.001316(9) |
| 5 | 0.025132 | 0.02514(3) | 0.000802 | 0.000804(6) |
| 6 | 0.017868 | 0.01787(2) | 0.000544 | 0.000543(4) |

Table 15: Numerical results for the regularized SU(3) Polyakov chain link model at the coupling $\beta = 0.5i$, $\kappa = 1$ and $\mu = 0.5$ with a regularization force $r = -25$ for the observables $\mathcal{O}_k = \left(\text{Tr}\left[P^{-1}\right]/3\right)^k$, $k = 1, \ldots, 6$. In contrast to the original model, the regularization term leads to correct convergence. The same notation as in Table 2 is used.

| Order $k$ | $\text{Re}\langle\mathcal{O}_k\rangle_{\rho+R}^{\text{exact}}$ | $\text{Re}\langle\mathcal{O}_k\rangle_{\rho+R}$ | $\text{Im}\langle\mathcal{O}_k\rangle_{\rho+R}^{\text{exact}}$ | $\text{Im}\langle\mathcal{O}_k\rangle_{\rho+R}$ |
|---|---|---|---|---|
| 1 | 0.258010 | 0.2580(1) | 0.007344 | 0.00736(6) |
| 2 | 0.115227 | 0.11522(7) | 0.004731 | 0.00472(2) |
| 3 | 0.071715 | 0.07171(5) | 0.002366 | 0.00237(1) |
| 4 | 0.039539 | 0.03955(3) | 0.001504 | 0.001506(9) |
| 5 | 0.025096 | 0.02510(3) | 0.001025 | 0.001025(6) |
| 6 | 0.017848 | 0.01785(2) | 0.000649 | 0.000653(4) |

Table 16: Bias-corrected results for the regularized SU(3) Polyakov chain link model at the coupling $\beta = 0.5i$, $\kappa = 1$ and $\mu = 0.5$ with a regularization force $r = -25$ for the observables $\mathcal{O}_k = (\text{Tr}[P]/3)^k$, $k = 1, \ldots, 6$. The bias correction procedure using the Dyson-Schwinger equation of $\text{Tr}[P]$ yields results that are in agreement with the exact results. The same notation as in Table 3 is used.

| Order $k$ | $\text{Re}\langle\mathcal{O}_k\rangle_{\rho}^{\text{exact}}$ | $\text{Re}\langle\mathcal{O}_k\rangle_{\rho+R}^{\text{bias-corr.}}$ | $\text{Im}\langle\mathcal{O}_k\rangle_{\rho}^{\text{exact}}$ | $\text{Im}\langle\mathcal{O}_k\rangle_{\rho+R}^{\text{bias-corr.}}$ |
|---|---|---|---|---|
| 1 | 0.347315 | 0.349(3) | 0.259843 | 0.258(2) |
| 2 | 0.152430 | 0.153(2) | 0.139517 | 0.1398(7) |
| 3 | 0.082942 | 0.083(1) | 0.097470 | 0.0976(7) |
| 4 | 0.047047 | 0.0475(8) | 0.070313 | 0.0701(5) |
| 5 | 0.027324 | 0.0277(6) | 0.047738 | 0.0476(5) |
| 6 | 0.017663 | 0.0179(4) | 0.035714 | 0.0356(4) |

Table 17: Bias-corrected results for the regularized SU(3) Polyakov chain link model at the coupling $\beta = 0.5i$, $\kappa = 1$ and $\mu = 0.5$ with a regularization force $r = -25$ for the observables $\mathcal{O}_k = \left(\text{Tr}\left[P^{-1}\right]/3\right)^k$, $k = 1, \ldots, 6$. The bias correction procedure using the Dyson-Schwinger equation of $\text{Tr}[P]$ yields results that are in agreement with the exact results. The same notation as in Table 3 is used.

| Order $k$ | $\text{Re}\langle\mathcal{O}_k\rangle_{\rho}^{\text{exact}}$ | $\text{Re}\langle\mathcal{O}_k\rangle_{\rho+R}^{\text{bias-corr.}}$ | $\text{Im}\langle\mathcal{O}_k\rangle_{\rho}^{\text{exact}}$ | $\text{Im}\langle\mathcal{O}_k\rangle_{\rho+R}^{\text{bias-corr.}}$ |
|---|---|---|---|---|
| 1 | 0.419999 | 0.422(4) | 0.156268 | 0.155(2) |
| 2 | 0.191396 | 0.192(2) | 0.156389 | 0.1562(5) |
| 3 | 0.096932 | 0.098(1) | 0.103684 | 0.1038(6) |
| 4 | 0.053733 | 0.054(1) | 0.069514 | 0.0693(5) |
| 5 | 0.032540 | 0.0329(6) | 0.051796 | 0.0516(4) |
| 6 | 0.020094 | 0.0205(5) | 0.037707 | 0.0376(4) |

Table 18: Bias-corrected results for the regularized SU(3) Polyakov chain link model at the coupling $\beta = 0.5i$, $\kappa = 1$ and $\mu = 0.5$ with a regularization force $r = -25$ for the observables $\mathcal{O}_k = (\text{Tr}[P]/3)^k$, $k = 1, \ldots, 6$. The bias correction procedure using the Dyson-Schwinger equation of $\text{Tr}[P^{-1}]$ yields results that are in agreement with the exact results. The same notation as in Table 3 is used.

| Order $k$ | $\text{Re}\langle\mathcal{O}_k\rangle_\rho^{\text{exact}}$ | $\text{Re}\langle\mathcal{O}_k\rangle_{\rho+R}^{\text{bias-corr.}}$ | $\text{Im}\langle\mathcal{O}_k\rangle_\rho^{\text{exact}}$ | $\text{Im}\langle\mathcal{O}_k\rangle_{\rho+R}^{\text{bias-corr.}}$ |
|---|---|---|---|---|
| 1 | 0.347315 | 0.347(2) | 0.259843 | 0.259(2) |
| 2 | 0.152430 | 0.1524(8) | 0.139517 | 0.140(1) |
| 3 | 0.082942 | 0.0827(8) | 0.097470 | 0.098(1) |
| 4 | 0.047047 | 0.0471(6) | 0.070313 | 0.0701(7) |
| 5 | 0.027324 | 0.0274(4) | 0.047738 | 0.0476(5) |
| 6 | 0.017663 | 0.0177(3) | 0.035714 | 0.0355(4) |

Table 19: Bias-corrected results for the regularized SU(3) Polyakov chain link model at the coupling $\beta = 0.5i$, $\kappa = 1$ and $\mu = 0.5$ with a regularization force $r = -25$ for the observables $\mathcal{O}_k = \left(\text{Tr}[P^{-1}]/3\right)^k$, $k = 1, \ldots, 6$. The bias correction procedure using the Dyson-Schwinger equation of $\text{Tr}[P^{-1}]$ yields results that are in agreement with the exact results. The same notation as in Table 3 is used.

| Order $k$ | $\text{Re}\langle\mathcal{O}_k\rangle_\rho^{\text{exact}}$ | $\text{Re}\langle\mathcal{O}_k\rangle_{\rho+R}^{\text{bias-corr.}}$ | $\text{Im}\langle\mathcal{O}_k\rangle_\rho^{\text{exact}}$ | $\text{Im}\langle\mathcal{O}_k\rangle_{\rho+R}^{\text{bias-corr.}}$ |
|---|---|---|---|---|
| 1 | 0.419999 | 0.4207(8) | 0.156268 | 0.1560(4) |
| 2 | 0.191396 | 0.191(1) | 0.156389 | 0.157(1) |
| 3 | 0.096932 | 0.0968(8) | 0.103684 | 0.1040(9) |
| 4 | 0.053733 | 0.0538(5) | 0.069514 | 0.0694(6) |
| 5 | 0.032540 | 0.0326(4) | 0.051796 | 0.0516(5) |
| 6 | 0.020094 | 0.0202(3) | 0.037707 | 0.0376(4) |

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
