# Peer review of "Lefschetz thimble-inspired weight regularizations for complex Langevin simulations"

_SciPost Physics, doi:SciPost Phys. 18, 092 (2025)_

## Round 1 · Referee Report · Rasmus Larsen (Referee 2) · 2025-2-7

Report

The authors responded to all my comments and I am happy with the changes.

I am also fine with the other changes done to the paper.

The acceptance criteria are still fulfilled as explained in the first report.
No change to the strength and weaknesses.

I therefore recommend the paper for publication.

Regards Rasmus Larsen

Recommendation

Publish (meets expectations and criteria for this Journal)

---

## Round 1 · Referee Report · Alexander Rothkopf (Referee 1) · 2025-2-17

Report

The authors have carefully addressed all points raised in my report and modified the text to my satisfaction. As laid out in the original report, the paper fulfils the acceptance criteria for SciPost Physics and with the changes made to the second version I can recommend the manuscript for publication.

Recommendation

Publish (meets expectations and criteria for this Journal)

---

## Round 1 · Author Response

SciPost Reports

We would like to thank the Referees for their careful reading, detailed reports, and very useful suggestions. We have addressed each of them in detail and modified the manuscript accordingly. Report 1

1) p2 i.e. Euclidean field theories -> that can be formulated solely using Euclidean time.

Thank you for the revised formulation, we have adapted the manuscript accordingly.

2) p.3 "there is an absence of general, constructive methods to achieve reliable convergence". Since the authors remark in the introduction paragraph that the sign problem is NP hard, it follows that such a general constructive method cannot exist, otherwise P=NP. We appreciate the Referee's insightful perspective. Our original formulation may not have been sufficiently clear, and we address their statement through the following three key points: First, by “general, constructive methods,” we did not intend to imply the existence of a method that entirely avoids the sign problem and thus solves an NP-hard problem. Rather, we aimed to highlight that it remains an open question of which specific features of a theory lead to incorrect convergence and how to address such issues. A deeper understanding of how to characterize models based on correct or incorrect convergence of CL is desirable but still lacking. Importantly, this would not necessarily resolve the sign problem. Moreover, methods that address incorrect convergence might still be model parameter-specific and, therefore, may not alleviate the exponential scaling of computational cost. From this perspective, this is not in conflict with the assumption that P ≠ NP. Second, incorrect convergence is not always correlated with the severity of the sign problem. While numerical instability often correlates with the intensity of the sign problem, there are cases where CL converges incorrectly even when the sign problem is mild. In our work, we focus primarily on the real probability density obtained via CL, which characterizes the complex weight function specific to the model. Since this also depends highly on model parameters (such as coupling, volume, etc.), we do not directly analyze how the computational time scales with respect to ‘model complexity’. Finally, even if we were to discover a general, constructive method for achieving the correct convergence of complex Langevin, this would not necessarily resolve the sign problem itself. The construction could still scale exponentially, and the computational cost of the “corrected” complex Langevin process could also grow exponentially. The ultimate goal has to be the development of a procedure that is better conditioned than the original sign problem. To acknowledge the Referee’s concern and avoid misunderstandings with the broader audience, we have rephrased the specific part of our manuscript.

3) p.4 perhaps it would be good for the reader unfamiliar with the work of Doi and Tsutsui to mention that one key limitation to their approach is that in their original formulation they need to compute the conventional reweighing factor and thus face the same difficulty as reweighing itself.

We thank the Referee for this suggestion. We have modified the second-to-last paragraph to emphasize the limitations of the original approach and our methodological improvement. We have also added a footnote on page 10 after the introduction of the weight regularization method and before the novel methods to compute the ratio Q.

4) p.8 The quantity in (19) to investigate the spread of the distribution of CL (boundary terms) is described as observable-independent. I do not agree with this terminology, as (19) chooses one particular observable and different observables can exhibit different tail structures in histograms. The difficulty with the boundary criterion is that no observable may exhibit a tail structure for CL to converge correctly. One may find some, whose histogram decays quickly but there may be others that still exhibit tails.

We thank the Referee for that interesting question. In the following, we hope we can clarify and justify why the description observable-independent is used in our work. We would like to emphasize that the drift magnitude density and its decay is a different criterion of correctness introduced by Nagata et al [arXiv:1606.07627]. The proof is based on the expansion of the time evolution operator of the Fokker-Planck equation. It was shown that if and only if the drift magnitude density decays at least exponentially fast, the expansion has a uniform (with respect to the Langevin time) lower bound for its convergence radius and hence can converge correctly. This proof is different than the criterion of correctness based on boundary terms shown in [arXiv:1808.05187]. In the same work, it was also remarked that the criterion by Nagata et al does not involve the measurement of observable. Hence, in our opinion the terminology ”observable-independent” is justified. As commented by the Referee: the downside of the boundary term criterion is that it needs to be checked for an observable. This is the reason why we checked the drift magnitude condition instead as it is easily accessible in all models considered in this work.

5) p.10 After a back of the envelope calculation, I am unsure about eq. (28). How do the authors manage to cancel the <O>\rho contribution from (24)? When subtracting <O>_r2 -<O>_r1 , the denominator comes with two different values of Q. After getting to a common denominator and moving that denominator to the LHS, the terms with <O>\rho seem to not trivially cancel. What have I overlooked there?

We thank the Referee for the detailed analysis of your calculations. As a first step, Eq. (24) can be rewritten to obtain Eq. (26). Note that the left-hand side of Eq. (26), <O>_\rho, is independent of the regularization parameter r as \rho is the weight function of the original model. We express \tilde{Q}, and arrive at Eq. (28), by setting the right-hand side equal for two different values r_1 and r_2.

6) p.12 Could the authors give a bit more details in how they arrive at choosing the particular regularization term. For the complex cosine model a similar regularization term was used by Doi and Tsutsui. Does this mean that this is the only viable regularization term or could one modify it?

The construction of a suitable regularization is highly dependent on the considered model and its corresponding weight function. Indeed, the regularization for the cosine model is inspired by the work of Tsutsui and Doi, as we commented in the second-to-last paragraph of Section 4.0. However, the design of the regularization term is not necessarily unique and we have found other successful weight regularizations for the theories, albeit we show only the most efficient ones.

We discuss the strategy of how we construct the weight regularizations in Sec. 3.2. We have now added a step-by-step recipe that we suggest and follow in our work of how suitable weight regularization can be constructed.

We emphasize that the stated procedure does not yield a unique regularization, but the existence of an adequate regularization seems likely. However, its proof is work for future developments.

7) p.13 (comment) While numerical evidence and intuitive arguments are given for the success of the method, it is worrying that the modification term proposed by the method leads to a non-holomorphic effective action. It puts the approach on a similar footing as the dynamical stabilisation approach, which too, while successful in practice does not allow a formal application of the proof of convergence.

We thank the author for this observation. We emphasize that this is only the case for the first regularization for the cosine model, all other regularizations studied in the work are holomorphic (up to isolated points which can be shown to have no effect on complex Langevin). It was further argued that the non-holomorphic points/line are weighted by zero measure and hence do not contribute. In contrast, the dynamical stabilization term is nowhere complex differentiable as it essentially depends on the Frobenius (unitarity) norm of the link variables and hence technically has to be extrapolated to vanishing force. In addition, all singularities in the complex plane were checked to be repulsive with respect to the complex Langevin drift and hence do not distort/contribute to the results. Furthermore, we have developed an analytical approach for how to correct for the bias that is introduced by the regularization. We therefore disagree with the assessment that the method is directly comparable to and stands on a similar footing as dynamical stabilization.

8) p.15 "can arise without any numerical signatures in the observables" -> "can arise without any numerical pathologies in the observables".

Thank you for the improved formulation, we adapted the manuscript accordingly.

9) p.19 reduced Polykov -> reduced Polyakov

We corrected the spelling error, thank you for pointing it out.

10) A legend for the colour coding in the scatter plots is missing

We thank the Referee for bringing this to our attention. Initially, we deliberately omitted color coding for the histogram, as the logarithmic scaling complicates quantitative analysis and was primarily chosen to highlight the histogram's decay. However, since this emphasis remains unaffected by the presence of color bars, we have now included them.

11) In the conclusion, could the authors rephrase the first paragraph describing the content of the study. I do not see that the authors have investigated the criterion for correct convergence of CL but have used it to confirm that a proposed modification strategy reproduces the correct results in several model systems.

We thank the Referee for his note. The authors would like to emphasize that the regularization technique is grounded in the structure of Lefschetz thimbles, with the regularization terms being directly motivated by this structure. Since the conjecture "one compact thimble implies correct convergence of the CL" lacks formal mathematical proof, we provide numerical evidence to support this conjecture throughout the work. In this context, we explore the correctness criterion and its relationship to the structure of the thimbles. Furthermore, in Appendix A, we show that when multiple compact thimbles are present, the criterion is not satisfied. We wish to stress that this formulation is intentional, as the main takeaway from our work is the connection between Lefschetz thimbles and the correctness criterion. However, we have revised the first paragraph to better incorporate the Referee’s perspective.

Report 2

1) below eq. 27 <O>_R = r <O>_G should be <O>_R = <O>_G as <O>_R = int dx OrG / int dx r*G so the r’s should cancel.

eq. 28 seems to agree with this assessment.

Thank you for pointing this out, we have corrected that mistake. Indeed, Eq. (28) correctly respects the independence of <O>_R of r when assuming R_0=0.

2) Figure 4, 10 and 14 left. What are the different colors? Would guess Re and Im, but was not shown anywhere (besides the right figures).

The authors have decided not to show the legend for the left plots in these figures as the right plots show the same quantities with the same color and include a legend specifying which color corresponds to the real and imaginary parts.

3) Is there a good reason for why r=-5 in section 5 and r=-25 in section 6? How were these values found? Why negative?

The regularization force parameter r is essentially found by scanning the real line. For a given parameter, we check if the Lefschetz thimbles of the corresponding resulting regularized model satisfy the desired properties: up to global symmetries such as point symmetries in the complex plane, only one compact thimble contributes to the path integral. This is the case for a range of values for r, and we chose it such that |r| is as small as possible as this minimizes the statistics needed to evaluate Eq. (29). The fact that they are negative is indeed curious but stems merely from the fact that we have found that for positive parameters the desired thimble structure was not reached, while it was obtained for negative values. This is due to the number, type, and position of stationary points in the complex plane.

4) There should be a minus for derivative of Tr[P^-1] in eq. 55 just as there correctly is for K_j in eq 52 as dTr(P^a)/dtau_j = iTr(atau_j*P^a) (ignoring the specific insertion position in the chain.)

We thank the Referee for looking into our calculations so carefully. Yes, indeed a minus sign is missing for the derivative of the inverse Polyakov chain. We have corrected the typo in the manuscript and we confirm that we have implemented the correct formula for our simulations.

Suggestions (feel free to ignore): 1s) eq. 30 is correct but does come a bit out of nowhere. A simple comment about it arising from the invariance of of the integral under a shift in a variable x_0 and it is the derivative with respect to x0 of <O> which is zero, that generates these equations, would make it easier to follow.

Thank you for that comment. To improve the clarity of where this equation comes from, we have added another equality to an integral over a total derivative, which vanishes identically.

---

## Round 1 · List of Changes

• Added colorbars for all histogram plots
  • Minor changes of formulations in paragraphs one, five, and eight of the first paragraph in Section 1
  • Added footnote on page 10
  • Changed Eq. (30) on page 11
  • Added paragraph at the end of Section 3.2
  • Minor changes of formulations in paragraph one of the first paragraph in Section 4.2
  • Fixed typo in Eq. (54) on page 25
  • The rephrased first paragraph of Section 7

---

## Editorial Decision

published